# A nuclear role for the DEAD-box protein Dbp5 in tRNA export

**Azra Lari[1], Arvind Arul Nambi Rajan[2], Rima Sandhu[3], Taylor Reiter[4], Rachel Montpetit[3], Barry P Young[5], Chris JR Loewen[5], Ben Montpetit[1,2,3,4]\***

[1]Department of Cell Biology, University of Alberta, Edmonton, Canada; [2]Biochemistry, Molecular, Cellular and Developmental Biology Graduate Group, University of California, Davis, Davis, United States; [3]Department of Viticulture and Enology, University of California, Davis, Davis, United States; [4]Food Science Graduate Group, University of California Davis, Davis, United States; [5]Department of Cellular and Physiological Sciences, Life Sciences Institute, University of British Columbia, Vancouver, Canada

**Abstract** Dbp5 is an essential DEAD-box protein that mediates nuclear mRNP export. Dbp5 also shuttles between nuclear and cytoplasmic compartments with reported roles in transcription, ribosomal subunit export, and translation; however, the mechanism(s) by which nucleocytoplasmic transport occurs and how Dbp5 specifically contributes to each of these processes remains unclear. Towards understanding the functions and transport of Dbp5 in *Saccharomyces cerevisiae*, alanine scanning mutagenesis was used to generate point mutants at all possible residues within a GFP-Dbp5 reporter. Characterization of the 456 viable mutants led to the identification of an N-terminal Xpo1-dependent nuclear export signal in Dbp5, in addition to other separation-of-function alleles, which together provide evidence that Dbp5 nuclear shuttling is not essential for mRNP export. Rather, disruptions in Dbp5 nucleocytoplasmic transport result in tRNA export defects, including changes in tRNA shuttling dynamics during recovery from nutrient stress.

**\*For correspondence:** benmontpetit@ucdavis.edu

**Competing interests:** The authors declare that no competing interests exist.

## Introduction

In eukaryotic cells, spatial separation of the transcriptional and translational processes necessitates transport of RNA and protein cargos across the nuclear envelope (NE) via nuclear pore complexes (NPCs). Generally, messenger RNAs (mRNAs) are transcribed and processed in the nucleus, transported across the NE, and then translated, stored, or decayed in the cytoplasm. Non-coding RNA (ncRNA) species, such as ribosomal RNA (rRNA) or transfer RNA (tRNA), also undergo nuclear processing and export following transcription (*Hopper, 2013*; *Zemp and Kutay, 2007*). Throughout these events, RNA-binding proteins (RBPs) interact with each RNA to form a ribonucleoprotein complex (RNP), with the protein constituents of each RNP mediating RNA processing and RNP functions within the cell (*Moore, 2005*).

DEAD-box proteins (DBPs) are a family of RBPs found in all kingdoms of life, often displaying RNA-stimulated ATPase activity, with diverse roles in RNA biology (*Linder and Jankowsky, 2011*). For example, DEAD-box protein 5 (Dbp5 or DDX19B in humans) plays an essential role in eukaryotic mRNA metabolism by driving directional nuclear mRNP export (*Hodge et al., 1999*; *Lund and Guthrie, 2005*; *Schmitt et al., 1999*; *Snay-Hodge et al., 1998*; *Tran et al., 2007*; *Tseng et al., 1998*). Dbp5, as a DEAD-box protein, is characteristically composed of two RecA-like domains that form the catalytic core of the enzyme, with identifiable sequence motifs that mediate RNA-binding, ATP-binding, and ATP hydrolysis (*Collins et al., 2009*; *Dossani et al., 2009*; *Fan et al., 2009*; *Montpetit et al., 2011*; *Napetschnig et al., 2009*; *von Moeller et al., 2009*). Dbp5 ATPase activity is further modulated by cytoplasmic oriented NPC proteins, Nup42, Nup159 (NUP214 in humans),

and Gle1 together with the endogenous small molecule co-factor, inositol hexakisphosphate (InsP$_6$) (*Adams et al., 2017*; *Alcázar-Román et al., 2006*; *Dossani et al., 2009*; *Hodge et al., 2011*; *Hodge et al., 1999*; *Montpetit et al., 2011*; *Napetschnig et al., 2009*; *Noble et al., 2011*; *Schmitt et al., 1999*; *von Moeller et al., 2009*; *Weirich et al., 2006*; *Weirich et al., 2004*; *Wong et al., 2018*).

Based on the localization of Dbp5 at NPCs and regulation by NPC components, models propose that Dbp5 encounters mRNPs at the cytoplasmic face of NPCs and remodels these substrates to enforce directionality as mRNPs enter the cytoplasm (*Folkmann et al., 2011*; *Heinrich et al., 2017*; *Stewart, 2007*). However, Dbp5 is dynamic at NPCs and is known to shuttle between the nuclear and cytoplasmic compartments (*Estruch and Cole, 2003*; *Hodge et al., 1999*; *Izawa et al., 2005*; *Noble et al., 2011*; *Takemura et al., 2004*; *Zhao et al., 2002*). This information prompts an alternative model in which Dbp5 functions as a scaffold, binding the mRNA in the nucleoplasm and traveling with nuclear mRNPs to NPCs where remodeling occurs through interactions with Gle1 and Nup159 (reviewed in *Heinrich et al., 2017*). While nuclear shuttling of Dbp5 is known, a mechanistic understanding of how nuclear transport is controlled and the relevance of this activity in the context of mRNP assembly and export is not.

Several additional roles for Dbp5 in gene expression have been proposed that point to functions of Dbp5 in the nuclear and cytoplasmic compartments. For example, studies have shown both physical and genetic interactions between Dbp5 and transcription initiation and translation termination machinery (*Alcázar-Román et al., 2010*; *Beißel et al., 2019*; *Bolger et al., 2008*; *Estruch et al., 2012*; *Estruch and Cole, 2003*; *Gross et al., 2007*). Additionally, a role for Dbp5 in the nuclear export of pre-ribosomal subunits has been identified (*Neumann et al., 2016*). Most recently, the mammalian and *Xenopus* homologs of Dbp5, DDX19B, have been shown to stabilize ribosomal elongation and termination complexes in vitro and contribute to nuclear R-loop clearance upon replication or DNA damage stress (*Hodroj et al., 2017*; *Mikhailova et al., 2017*). These data suggest a broader role for Dbp5 in RNA metabolism and gene expression, but the mechanisms by which Dbp5 contributes to these processes remains undefined.

With the goal of understanding the role of Dbp5 within the nuclear compartment, a comprehensive alanine scanning mutagenesis approach was employed to generate mutants encompassing residues 2–482 of Dbp5. The resulting data provide a functional map of Dbp5 at a single amino acid resolution and mutant alleles that can be employed to detail mechanisms by which Dbp5 contributes to gene expression. This is exemplified by work here describing the identification of single mutations within Dbp5 that facilitate nuclear shuttling of Dbp5 and the discovery of a novel role for Dbp5 in nuclear tRNA export.

## Results

### Characterization of the *dbp5* mutant collection

An alanine scanning mutagenesis approach was used to generate single alanine (or glycine if the residue was an alanine) substitutions at positions 2–482 of Dbp5, resulting in a collection of 481 plasmids carrying *dbp5* mutant alleles in frame with GFP at the N-terminus. To begin to address the functional status of each mutant, a plasmid shuffle assay was used to generate strains solely expressing the mutated version of GFP-Dbp5. Based on this selection, 25 alanine substitutions were identified as lethal (*Supplementary file 1* - Table 1). Consistent with published biochemical and structural models of DBPs (*Linder, 2006*), the majority of the lethal substitutions identified in Dbp5 were in motifs known to be critical for RNA-binding, ATP-binding, and ATP hydrolysis (*Figure 1—figure supplement 1*). For example, lethal mutations included *Q119A*, which is within the highly conserved Q-motif and is known to be required for adenine recognition and ATP hydrolysis (*Cordin et al., 2004*). Similarly, lethal mutations included *D239A*, *E240A*, and *D242A* in motif II (or the Walker B motif), which are part of the namesake D-E-A-D amino acid sequence critical for ATP recognition and hydrolysis. Lethal mutations also mapped to regions of Dbp5 that participate in binding NPC regulators Gle1 and Nup159, including residues R256, Y325, and K382 (*Dossani et al., 2009*; *Montpetit et al., 2011*; *Weirich et al., 2004*). Further phenotypic characterization of the individual point mutants within this collection is underway and will be published elsewhere, the focus of this work is on the nuclear transport of Dbp5.

## Dbp5 contains an N-terminal nuclear localization signal

The inclusion of GFP in the mutant collection allowed for Dbp5 localization to be assayed in each point mutant. Under steady-state growth conditions, in the presence of a wild-type allele, GFP-Dbp5 expression was detectable for all 481-point mutations. The majority of the mutants displayed the expected localization pattern with enrichment of Dbp5 at NPCs with diffuse cytoplasmic and nuclear pools (*Schmitt et al., 1999*; *Snay-Hodge et al., 1998*; *Weirich et al., 2004*); however, two strains (GFP-*dbp5-L12A and GFP-dbp5-L15A*) showed a prominent nucleoplasmic localization of GFP-Dbp5 (*Figure 1A*). In these mutants, GFP-Dbp5 was found both in the nucleoplasm and nucleolus, as shown through co-localization of GFP-Dbp5$^{L12A}$ with the nucleolar protein Nop1 (*Figure 1A*). As sole copy, a plasmid based GFP-*dbp5-L12A* allele displayed a temperature sensitive growth and poly(A)-RNA export defect at 37°C, but upon removal of GFP and integration of *dbp5-L12A* into the genome at the endogenous gene locus, these phenotypes were no longer present (*Figure 1—figure supplement 2A–B*). These data indicated that N-terminal GFP tagging in the context of the L12A mutation caused growth and mRNP export defects at 37°C. This interpretation is based on the fact that a strain with the wild-type *DBP5* allele did not show these defects when tagged with GFP and grown at 37°C (*Figure 1—figure supplement 2A–B*). The integrated and untagged version of *dbp5-L12A* was also assessed by immunofluorescence (IF), and while the NPC associated pool of Dbp5 was not retained in wild-type during the IF procedure, Dbp5$^{L12A}$ was enriched in the nucleoplasm (*Figure 1—figure supplement 2C*). This data shows that the presence of GFP did not contribute to the nuclear localization of Dbp5$^{L12A}$. Given the strong nuclear localization of Dbp5$^{L12A}$, plus the previous reports of Dbp5 functioning in translation control (*Beißel et al., 2019*; *Bolger et al., 2008*; *Gross et al., 2007*), the overall fitness of a strain carrying an untagged and integrated allele of *dbp5-L12A* raises questions as to the requirements for Dbp5 in the cytoplasm, which this allele could be used to address.

Residues surrounding L12 and L15 are hydrophobic and follow a consensus pattern similar to a nuclear export signal (NES) sequence (*Figure 1B*) (*la Cour et al., 2004*). To determine if this region was sufficient to function as an NES, residues 1 to 52 of Dbp5 (NES$^{DBP5}$) were appended to a 2xGFP reporter with an SV40 nuclear localization signal sequence (NLS$^{SV40}$) to generate a 2xGFP-NLS$^{SV40}$-NES$^{DBP5}$ reporter. The addition of NES$^{DBP5}$ antagonized NLS$^{SV40}$ activity and facilitated export of the 2xGFP-NLS$^{SV40}$-NES$^{DBP5}$ reporter in a manner comparable to a prototypical PKI NES (*Fischer et al., 1995*; *Wen et al., 1995*). Activity of the NES$^{DBP5}$ was abolished by the *L12A* substitution consistent with the impact of the L12A mutation in Dbp5 (*Figure 1B*). To test if export activity was dependent on Xpo1, the major karyopherin involved in nuclear export, the same assay was repeated in strains expressing Xpo1$^{T539C}$, which can be inhibited by Leptomycin B (LMB) treatment (*Maurer et al., 2001*; *Neville and Rosbash, 1999*). Upon addition of LMB, the GFP-NLS$^{SV40}$-NES$^{DBP5}$ reporter accumulated in the nucleus (*Figure 1C*), which is consistent with previous work showing that Dbp5 accumulated in the nucleus of a temperature sensitive (Ts) *xpo1-1* strain (*Hodge et al., 1999*). Together, these data show that the N-terminus of Dbp5 harbors an NES signal, which is sufficient to function as an Xpo1-dependent NES.

Given the essential function of Dbp5 at the cytoplasmic face of NPCs for mRNP export, a small and/or dynamic pool of Dbp5$^{L12A}$ must fulfill essential activities at NPCs. To test for the possibility that GFP-Dbp5$^{L12A}$ still accesses the cytoplasm, an anchor-away system was used with a peroxisomal anchor, Pex25-FKBP12, to anchor GFP-FRB-Dbp5 (present as sole copy of Dbp5) to peroxisomes in a rapamycin-dependent manner (*Haruki et al., 2008*). In wild-type cells, GFP-FRB-Dbp5 rapidly accumulated on peroxisomes after rapamycin addition, resulting in strong depletion of the GFP signal from the NE within ~4 minutes (*Figure 1D*). Similarly, GFP-FRB-Dbp5$^{L12A}$ was rapidly depleted from the nucleoplasm and accumulated on peroxisomes in the same time frame (data not shown). Dbp5 with alanine substitutions at L12, L15, and I17, referred to as GFP-FRB-*dbp5ΔNES*, also showed similar dynamics (*Figure 1D*), suggesting that the shuttling observed for GFP-FRB-Dbp5$^{L12A}$ was not due to residual NES activity. Quantitation of the nuclear GFP-FRB-Dbp5$^{ΔNES}$ showed that the nuclear signal was 63 ± 9% and 58 ± 7% of starting levels at 2 and 4 minutes after rapamycin treatment, indicating that GFP-FRB-Dbp5$^{ΔNES}$ shuttles and remains able to access the cytoplasm. This assay was also performed with wild-type GFP-FRB-Dbp5 in the background of an *xpo1-T539C* strain. Following treatment with LMB for ~7 minutes to allow GFP-FRB-Dbp5 to accumulate in the nucleus, rapamycin was added, but in this scenario GFP-FRB-Dbp5 remained largely nucleoplasmic

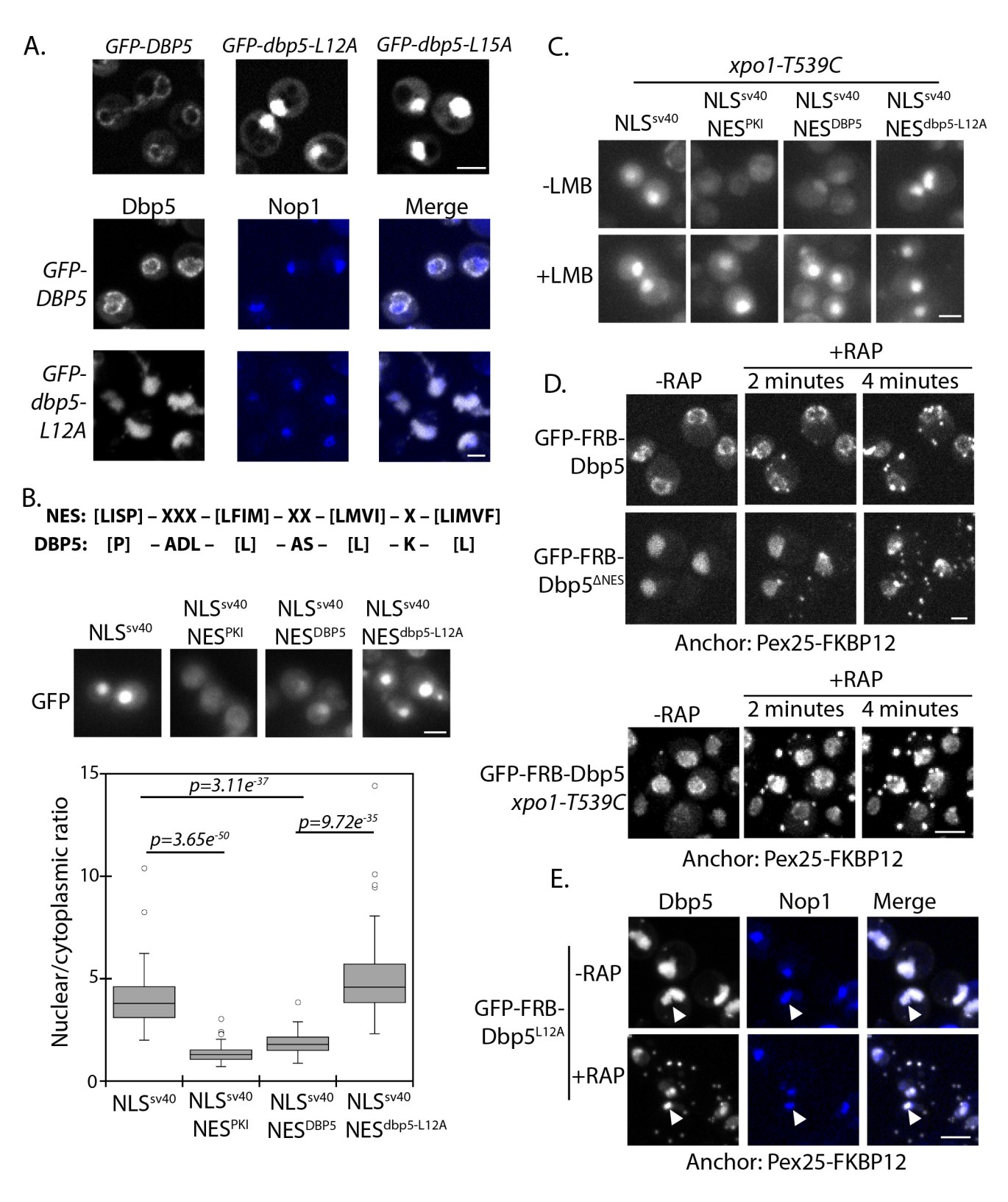

**Figure 1.** Identification of N-terminal nuclear export signal in Dbp5. (**A**) Fluorescent images showing localization of GFP-Dbp5 in plasmid based *GFP-DBP5, GFP-dbp5-L12A,* and *GFP-dbp5-L15A* strains (top panel). Bottom panel shows GFP-Dbp5 (gray) co-localization with the nucleolar marker Nop1-

*Figure 1 continued on next page*

*Figure 1 continued*

RFP (blue). (B) Schematic at the top shows the overall sequence composition of verified NES motifs (*la Cour et al., 2004*), as compared to the Dbp5 N-terminal amino acid sequence. Fluorescent images show localization of 2xGFP-NLS$^{SV40}$-NES$^{DBP5}$ reporters fused to residues 1–52 of *DBP5* or *dbp5-L12A* compared to a bona-fide NES (NES$^{PKI}$). Graph at the bottom shows the ratio between nuclear and cytoplasmic GFP signals measured for the various reporter constructs (n ≥ 100, error bars represent standard deviation, p-value from unpaired t-test with two-tailed distribution shown). (C) Fluorescent images show 2xGFP reporters as used in panel B in a strain carrying *xpo1-T539C* before and after addition of 100 ng/ml of LMB for 30 min to disrupt Xpo1 mediated export. (D) Fluorescent images show localization of GFP-FRB-Dbp5 or GFP-FRB-Dbp5$^{ΔNES}$ in cells with a Pex25-FKBP12 anchor prior to and following addition of 1 μg/ml rapamycin at 2 and 4 min. Bottom panel shows GFP-FRB-Dbp5 in cells with a Pex25-FKBP12 anchor and the *xpo1-539C* mutation to allow disruption of Xpo1 mediated export with LMB. Imaging was performed after treatment with 100 ng/ml of LMB for 7 min and following addition of rapamycin at 2 and 4 min. (E) Fluorescent images show localization of GFP-FRB-Dbp5$^{L12A}$ (gray) in reference to Nop1-RFP (blue) in cells with a Pex25-FKBP12 anchor before and after addition of rapamycin for 10 min. Co-localization of GFP-FRB-Dbp5$^{L12A}$ and Nop1-RFP indicated by white arrows. Scale bars = 2 μm. See *Figure 1—figure supplement 1* for primary Dbp5 sequence with position of DEAD-box motifs and lethal mutations. See *Figure 1—figure supplement 2* for further characterization of the *dbp5-L12A* allele.

The online version of this article includes the following figure supplement(s) for figure 1:

**Figure supplement 1.** DEAD-box motifs and lethal *dbp5* alleles.
**Figure supplement 2.** GFP-tagging of *dbp5-L12A* alters growth and mRNP export phenotypes.

(*Figure 1D*, bottom panel). Levels in the nucleoplasm measured 100 ± 23%, 99.5 ± 24%, and 103 ± 18% at 2, 4, and 10 minutes after rapamycin addition as compared to starting levels prior to rapamycin addition. These results show that Dbp5 export dynamics are Xpo1-dependent. As such, Dbp5 must engage the Xpo1 export pathway through both the N-terminal NES and via an independent means, likely in complex with other factors that also engage the Xpo1 transport system. It was also observed that a pool of GFP-FRB-Dbp5$^{ΔNES}$ remained in the nucleus of most cells following rapamycin addition, and based on co-localization of GFP-FRB-Dbp5$^{L12A}$ with Nop1, this pool of Dbp5 was associated with the nucleolus (*Figure 1E*). These data show that there are at least two distinct nuclear populations of GFP-FRB-Dbp5$^{L12A}$; one population that is dynamic and enters the cytoplasm through a mechanism dependent on Xpo1, and a second population that is stably associated with the nucleolus.

## Dbp5$^{R423A}$ shows altered nuclear import

To further identify mutants with altered Dbp5 nuclear transport, localization of GFP-Dbp5 was determined for the complete mutant collection in the background of a wild-type copy of Dbp5 following exposure to 12% ethanol. Acute ethanol shock is a condition whereby Dbp5 re-localizes to the nucleoplasm (*Izawa et al., 2005*; *Takemura et al., 2004*) and was used here as a tool to address Dbp5 nuclear transport. Using this assay, mutations in a stretch of residues within and near motif VI (422-429) were found to disrupt GFP-Dbp5 nuclear accumulation in ethanol (*Figure 2—figure supplement 1A*). Of these, the GFP-*dbp5-R423A* mutant was viable, displayed no growth defects at 25°C or 30°C, and was slow growing at 37°C (*Figure 2—figure supplement 1B*). In contrast, other mutations within motif VI that failed to relocalize to the nucleus were lethal (e.g. R426A and R429A). As observed for *dbp5-L12A*, GFP-*dbp5-R423A* showed a poly(A)-RNA export defect at 37°C, but upon removal of GFP and integration of the mutation into the genome, the *dbp5-R423A* strain grew comparably to wild-type at all temperatures and had no nuclear poly(A)-RNA accumulation phenotype (*Figure 2—figure supplement 1B–C*).

The accumulation of Dbp5 in the nucleoplasm of ethanol-treated cells has been reported to result from a disruption in nuclear export via Xpo1 (*Takemura et al., 2004*). Given the dependence of Dbp5 on Xpo1 for nuclear export, this suggested that failure of GFP-Dbp5$^{R423A}$ to accumulate in the nucleus during ethanol stress resulted from a defect in nuclear import. The region surrounding R423 is similar in sequence and charge to known NLS sequences (*Kosugi et al., 2009*); however, in isolation this region of Dbp5 when fused to 2xGFP did not support nuclear import of the reporter (data not shown). This suggests that other regions of Dbp5 may act in concert with R423 to mediate import, or the mutation of this residue may alter intra- or inter-molecular Dbp5 interactions that impact nuclear import efficiency.

To assay Dbp5$^{R423A}$ nuclear import by another means, the anchor-away technique was employed with a histone protein anchor (Htb2-FKBP12) to address nuclear access of wild-type or R423A versions of GFP-FRB-Dbp5 when present as sole copy. Upon addition of rapamycin, GFP-FRB-Dbp5

moved to the nucleoplasm, resulting in complete depletion of the GFP signal at the NE within ~11 minutes (*Figure 2B*). GFP-FRB-Dbp5[R423A] also accumulated in the nucleus following rapamycin addition, but a significant proportion remained apparent at the NE for the entire length of the 18 minute imaging series (*Figure 2B*). To test for the possibility that GFP-FRB-Dbp5[R423A] was stably bound to NPCs, the anchor-away system was again employed with the peroxisomal anchor (Pex25-FKBP12). In both wild-type cells and *dbp5-R423A* strains, GFP-FRB-Dbp5 rapidly accumulated on peroxisomes after rapamycin addition, resulting in complete depletion of the GFP signal from the NE in both strains in ~4 minutes (*Figure 2C*). Note the nuclear accumulation defect seen here was not as severe as compared with ethanol (see *Figure 2A*). A major difference between these two assays being the presence of a wild-type copy of Dbp5 in the ethanol experiments. Through use of both plasmid and integrated versions of Dbp5, it is observed that the presence of wild-type Dbp5 exacerbates the Dbp5[R423A] nuclear import defect (data not shown), which is suggestive of competition for nuclear transport. These results show that GFP-FRB-Dbp5[R423A] remains dynamic in the cytoplasm, but Dbp5[R423A] import efficiency is altered by this mutation.

Overall, the altered import kinetics of the *dbp5-R423A* allele and discovery of an N-terminal NES within Dbp5 provides a means to investigate the mechanics and functional importance of Dbp5 nuclear shuttling. Markedly, the lack of poly(A)-RNA accumulation in integrated versions of *dbp5-L12A,* where Dbp5 is predominantly nucleoplasmic, or *dbp5-R423A,* where Dbp5 nuclear access is altered, suggests that neither a large cytoplasmic pool of Dbp5 or rapid shuttling of Dbp5 through the nucleus is required to maintain steady-state nuclear mRNP export.

### *dbp5-L12A* and *dbp5-R423A* genetic interaction profiles

Towards understanding the functional consequence of biasing Dbp5 localization to the nucleus (i.e. *dbp5-L12A*) or altering nuclear access (i.e. *dbp5-R423A*), a synthetic gene array (SGA) screen was carried out to identify potential genetic interactions between these alleles and non-essential gene mutants. SGA screens have been carried out previously with the *DBP5* Ts allele *rat8-2*, yielding 108 potential synthetic interactions, encompassing proteins involved in chromatin remodeling, transcription, mRNA metabolism, and mRNP export (*Scarcelli et al., 2008*). In contrast, the screen with *dbp5-L12A* only found 12 potential synthetic interactions (*Figure 3A* and *Supplementary file 1* - Table 2), which included the exosome-associated component LRP1 and the TREX-2 complex component THP1. No gene ontology terms were enriched within this gene set, and most notably, genetic interactions were not identified with the many non-essential components of the mRNP export pathway, NPCs, or translation machinery. These results support the conclusion that having the majority of Dbp5 in the nucleus at steady-state is not detrimental to the process of mRNP export, translation, or overall cellular fitness.

The SGA screen with *dbp5-R423A* yielded 176 potential genetic interactions that were more consistent with the previously reported SGA screen (*Scarcelli et al., 2008*), including interactions with non-essential components of NPCs (e.g. MLP1, NUP2, NUP42, NUP60, NUP84, NUP120, NUP133 and POM33) and both mRNA and ncRNA export pathways (e.g. MSN5, NPL3, SAC3, SEM1, SLX9, THP1, and THP2) (*Figure 3* and *Supplementary file 1* - Table 2). These genetic interactions, combined with the observed poly(A)-RNA export defect only upon GFP tagging and growth at high temperature, suggest that when combined with other perturbations Dbp5[R423A] activity can be limiting for RNA export. An idea supported by the fact that a threshold level of Dbp5 ATPase activity has been determined to be required for mRNP export (*Dossani et al., 2009*). Indeed, the location of residue R423 in motif VI, a DEAD-box motif generally involved in nucleotide binding and hydrolysis, would be expected to alter enzymatic activity (*Hilbert et al., 2009*; *Linder and Jankowsky, 2011*). These facts prompted investigations to determine how the R423A substitution impacts ATPase activity, which may lead to synthetic interactions in the presence of other perturbations to the gene expression apparatus (e.g. deletions impacting the mRNP export apparatus) or the function of Dbp5 (e.g. N-terminal GFP tag). Using an in vitro ATPase assay, the basal steady-state ATPase rate of Dbp5[R423A] was found to be similar to wild-type, but in the presence of RNA, Dbp5[R423A] exhibited an activity $47 \pm 2\%$ of wild-type (*Figure 3B*). These results were the same at 37°C with Dbp5[R423A] RNA-stimulated ATPase activity at $52 \pm 7\%$ of wild-type. It is expected that the inability of Dbp5[R423A] to be fully stimulated by RNA results in the observed genetic interactions with other gene mutations that perturb RNP export and gene expression.

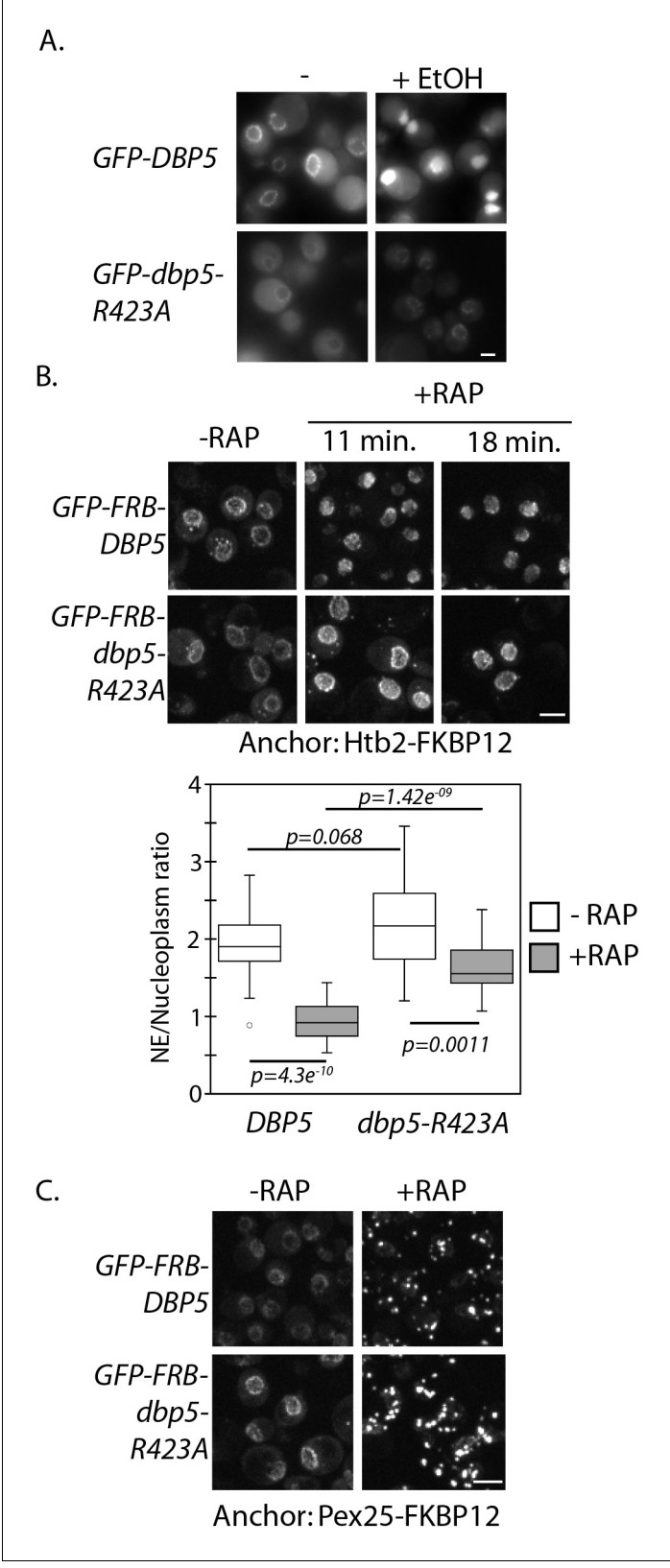

**Figure 2.** Altered import kinetics of Dbp5[R423A]. (**A**) Fluorescent images showing localization of plasmid expressed GFP-Dbp5 in *GFP-DBP5* and *GFP-dbp5-R423A* strains pre- and post-shift to media with 12% ethanol for 30 min. (**B**) Fluorescent images showing localization of GFP-FRB-Dbp5 or GFP-FRB-Dbp5[R423A] in cells with the nuclear

*Figure 2 continued on next page*

*Figure 2 continued*

Htb2-FKBP12 anchor following addition of 1 µg/ml rapamycin at the indicated time points. Graph shows quantification of the nuclear envelope signal intensity vs. nucleoplasm (n = 23, error bars indicate standard deviation, p-value from paired or unpaired t-test with two-tailed distribution shown). (C) Fluorescent images showing localization of GFP-FRB-Dbp5 or GFP-FRB-Dbp5[R423A] in cells with the Pex25-FKBP12 anchor following addition of 1 µg/ml rapamycin for 4 min. Scale bars = 2 µm. See *Figure 2—figure supplement 1* for characterization of other motif VI plasmid expressed GFP-Dbp5 mutants in ethanol.

The online version of this article includes the following figure supplement(s) for figure 2:

**Figure supplement 1.** Altered nuclear import kinetics of *dbp5* mutants in response to ethanol stress.

## Dbp5 functions in tRNA export within the nucleus

Recent work has linked Dbp5 to the export of ncRNAs, including ribosomal and telomerase RNAs (*Neumann et al., 2016*; *Wu et al., 2014*). Here, SGA screens have identified potential genetic interactions with factors that participate in rRNA (i.e. SLX9 and LRP1) and tRNA (i.e. MSN5 and MSW1) processing and export (*Figure 3* and *Supplementary file 1* - Table 2). Related to this, the Dbp5 trans-activator Nup159 and the mRNP export factor Mex67 were recently identified in a screen for proteins involved in nuclear tRNA biogenesis and export (*Wu et al., 2015*). Given these facts, rRNA processing and export were investigated in both the *dbp5-L12A* and *dbp5-R423A* mutants by northern blotting, rRNA FISH, and by use of rRNA export reporters; however, no evidence of rRNA processing or export defects were identified in either mutant (data not shown).

In the case of tRNAs, primary tRNA transcript processing involves removal of 5′ and 3′ ends of the tRNA in the nucleus, transport across the nuclear envelope, and splicing of intron containing tRNAs in the cytoplasm to produce functionally mature tRNAs (*Hopper, 2013*). The aberrant accumulation of tRNA intermediates signifies a breakdown in tRNA processing and/or export. To detect such intermediates, northern blotting was performed with a probe against the tRNA$^{Ile}_{UAU}$ in strains carrying integrated and untagged versions of wild-type, *dbp5-L12A*, and *dbp5-R423A*. The probe spans the 5′ exon and first 30 nucleotides of the tRNA$^{Ile}_{UAU}$ intron to allow detection of the primary transcript and processing intermediates. The ratio of intron containing intermediate (I) to precursor transcript (P) normalized to wild-type is used as a measure of tRNA processing, with altered ratios indicating a defect (*Wu et al., 2015*). At 25°C, the ratio between the precursor and intron containing tRNA in *dbp5-L12A* (1.00 ± 0.1) was unchanged from wild-type, while a small but consistent accumulation of the intron-containing tRNA intermediate was observed in *dbp5-R423A* (1.4 ± 0.1) (*Figure 4A*). This defect was further increased after two hours at 37°C in *dbp5-R423A* (2.3 ± 0.2), while the ratio only increased slightly in *dbp5-L12A* (1.4 ± 0.2) (*Figure 4A*). To verify that these defects went beyond a single tRNA species, northern blotting was also carried out with a probe against tRNA$^{Tyr}_{GUA}$, which also showed an accumulation of the intron containing species in *dbp5-R423A* (1.7 ± 0.4) at 37°C (*Figure 4B*). These measures suggest that tRNA processing or export is perturbed in the *dbp5-R423A* mutant.

Intron-containing tRNAs require transport to the cytoplasm to be spliced in yeast, therefore accumulation of intron containing intermediates in *dbp5-R423A* by northern blotting are suggestive of a defect in tRNA export (*Hopper, 2013*).Using a probe that binds all forms of tRNA$^{Ile}_{UAU}$, a significantly increased amount of tRNA$^{Ile}_{UAU}$ was observed in the nucleus of the integrated *dbp5-R423A* strain at 37°C by FISH, as compared to wild-type and *dbp5-L12A* strains (*Figure 4C*). Note that under these conditions and using integrated untagged alleles, no poly(A)-RNA export is observed in either *dbp5-L12A* or *dbp5-R423A*. The export defect in *dbp5-R423A* was similar to the level seen in the *mex67-5* strain, which has been previously identified to disrupt tRNA export (*Chatterjee et al., 2017*).

Dbp5[R423A] has reduced ATPase activity in the presence of RNA and altered nuclear access, either or both of which could influence tRNA export. To address this issue, the double mutant *dbp5-L12A/R423A* was generated with the idea that the L12A mutation would decrease the amount of Dbp5 in the cytoplasm and increase the amount in the nucleus. As a result, if the R423A mutation caused tRNA export defects by limiting Dbp5 ATPase activity in the cytoplasm, the L12A mutation would be expected to worsen this phenotype by reducing cytoplasmic levels of Dbp5. Alternatively, if Dbp5 nuclear access and activity within the nucleus is required to support tRNA export, the L12A mutation

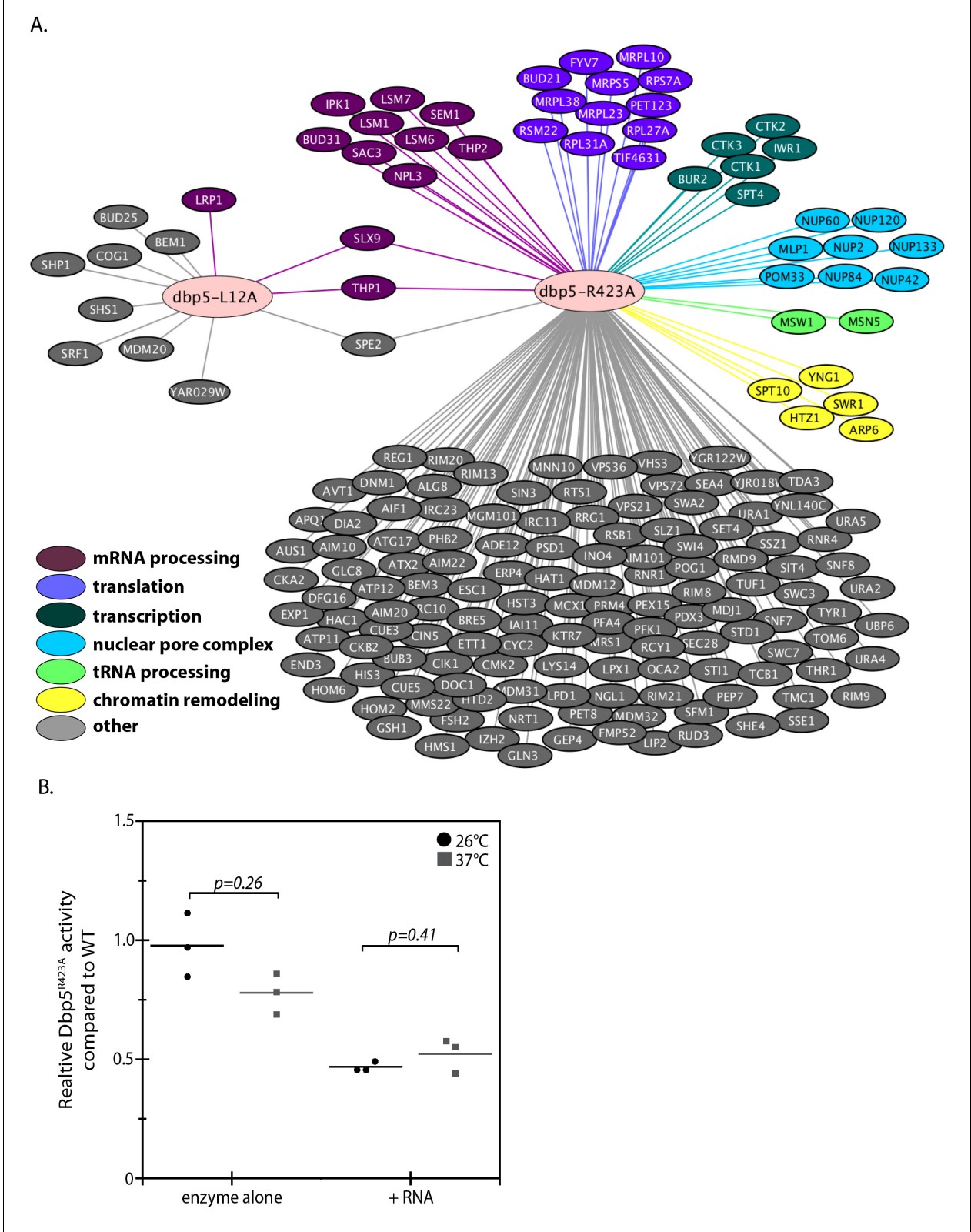

**Figure 3.** Genetic interaction analyses of *dbp5-L12A and dbp5-R423A.* (**A**) Identified synthetic interactions by SGA analysis in integrated and untagged *dbp5-L12A* and *dbp5-R423A* strains. Gene deletions are grouped based on gene ontology (magenta- mRNA processing, purple- translation, teal-transcription, blue- nuclear pore complex, green- tRNA processing, yellow- chromatin remodeling, gray- other). (**B**) Steady state in vitro ATPase assays

*Figure 3 continued*

performed with purified Dbp5 and Dbp5^R423A at 26°C or 37°C. Graph shows the average ATPase rate measured for Dbp5^R423A as a percentage of Dbp5 activity, with or without RNA stimulation (n = 3, error bars represent standard deviation, p-value from paired t-test with two-tailed distribution shown).

could improve defects in *dbp5-R423A* by increasing the amount of Dbp5 in the nuclear compartment. A strain expressing GFP-Dbp5^L12A/R423A is viable and there is an increased nuclear pool of Dbp5 as compared to R423A (*Figure 4—figure supplement 1A*); however, the untagged and integrated version of *dbp5-L12A/R423A* exhibited mRNP export defects at both 25°C and 37°C, and a strong temperature-dependent growth defect at 37°C (*Figure 4—figure supplement 1B–C*). This result indicates that further limiting the pool of cytoplasmic Dbp5 in the context of the R423A mutation alters mRNP export. Using the double mutant, northern blotting was performed with probes against the tRNA^Ile_UAU at 37°C (*Figure 4D*). As compared to *dbp5-R423A* (1.5 ± 0.2), accumulation of the intron containing tRNA intermediate was similar in the untagged and integrated *dbp5-L12A/R423A* (1.9 ± 0.5), suggesting that depleting the cytoplasmic pool of Dbp5 through inclusion of the L12A mutation did not further impact tRNA processing, in contrast to what is seen with mRNP export. Due to the presence of mRNP export defects at 25°C, this mutant was not used to further interrogate the role of Dbp5 in tRNA export. Combined, the SGA, northern, and in situ data employing the *dbp5-L12A* and *dbp5-R423A* mutations to alter Dbp5 nuclear levels are strongly supportive of Dbp5 functioning in tRNA export within the nucleus, while facilitating mRNP export in the cytoplasmic compartment (e.g. at the cytoplasmic face of NPCs).

## Gle1 is required for tRNA processing

Reported defects in tRNA processing in *nup159-1* (*Wu et al., 2015*), which is shared with *dbp5-R423A*, suggest a possible relationship between Dbp5 regulation at NPCs and Dbp5 functions in tRNA export. This prompted investigation of Gle1 in tRNA processing, given that Gle1 is also an NPC-associated regulator of Dbp5. In these assays, *dbp5-1* was included, since upon temperature shift this mutant has a strong mRNP export block similar to *nup159-1* and *gle1-4* (*Gorsch et al., 1995*; *Murphy and Wente, 1996*; *Tseng et al., 1998*), as does *mex67-5* (*Segref et al., 1997*). By northern blotting, the *dbp5-1* and *gle1-4* Ts mutants had a strong tRNA^Ile_UAU processing defect at 37°C that was comparable to *nup159-1* and *mex67-5* (*Figure 4—figure supplement 1D*). tRNA processing defects were also apparent in all four mutants at 25°C, which is a condition where mRNP export is not severely perturbed. These data demonstrate that Dbp5 and both regulators linked to Dbp5, Nup159 and Gle1, support tRNA processing and/or export as does the RNA export adaptor Mex67 (*Chatterjee et al., 2017*), under conditions that do not involve a strong mRNP export block (i.e. growth at 25°C).

Given the known role for these proteins in mRNP export and the function of Gle1 and Nup159 at NPCs with Dbp5, the localization of Dbp5 was considered in *gle1-4* and *nup159-1* as a possible cause for a shared tRNA processing defect. The localization of GFP-Dbp5^L12A was assayed at 25°C and at the same time point used for northern blotting (i.e. 2 hr at 37°C) leveraging the strong nuclear signal of GFP-Dbp5^L12A to assess Dbp5 nuclear localization. Under these conditions, it was observed that nuclear GFP-Dbp5^L12A localization was only reduced in *gle1-4* at 37°C, but remained largely unchanged in *gle1-4* at 25°C or *nup159-1* and *mex67-5* at 25°C or 37°C (*Figure 4—figure supplement 1E*). These data suggest that Gle1 and Nup159 influence tRNA export through a mechanism that goes beyond altering Dbp5 nuclear access.

## Dbp5 supports re-export of mature tRNAs following nutritional stress

It is known that re-export of mature tRNAs to the cytoplasm in response to nutrient availability involves the β-importin family member Exportin-5 (yeast Msn5) (*Bohnsack et al., 2002*; *Calado et al., 2002*; *Murthi et al., 2010*; *Takano et al., 2015*). The *msn5Δ/dbp5-R423A* double mutant was identified here as being synthetic sick by SGA (*Supplementary file 1* - Table 2), which is suggestive of a potential relationship between Dbp5 and tRNA re-export. A well characterized stress causing rapid and reversible re-localization of mature tRNAs to the nucleus is acute nutritional starvation using tRNA^Tyr_GUA (*Whitney et al., 2007*). To determine if regulated tRNA shuttling was altered in *dbp5-R423A*, tRNA^Tyr_GUA localization was assessed in strains subjected to a ten-minute

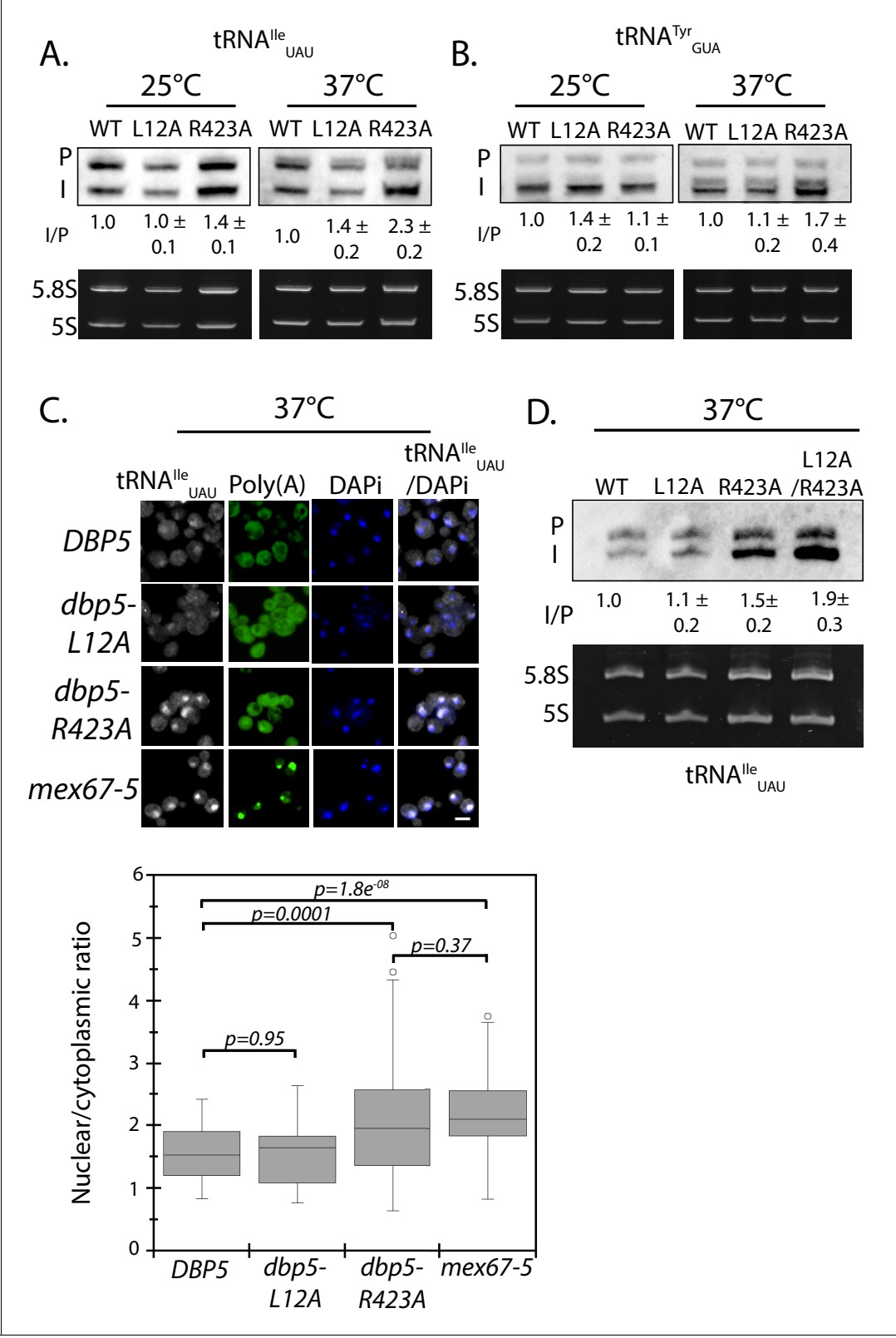

**Figure 4.** Dbp5 is required for tRNA export. (**A**) Northern blot analysis of tRNA$^{Ile}_{UAU}$ (Probe1) in integrated and untagged *DBP5 (WT)*, *dbp5-L12A*, and *dbp5-R423A* strains before and after a 2 hr temperature shift to 37°C. The primary tRNA transcript is denoted as (P), and the end-matured intron-containing tRNA is denoted as (I). Ratio of mean integrated intensity between (I) and (P) species is measured and normalized to the wild type value (n = 3, error represents standard deviation). Ethidium bromide stained gel shows 5.8S and 5S rRNA species in bottom panel as a loading control. (**B**)
*Figure 4 continued on next page*

*Figure 4 continued*

Northern blot analysis as in panel A using tRNA$^{Tyr}_{GUA}$ (Probe KC031). (**C**) Localization of tRNA$^{Ile}_{UAU}$ determined by FISH in integrated and untagged *dbp5-L12A* and *dbp5-R423A* strains after a 4 hr shift to 37°C. Cells were probed with a Cy3 end-labeled tRNA$^{Ile}_{UAU}$ probe (SRIM04, gray) and DAPi (blue). Scale bar = 2 μm. Graph shows the ratio between nuclear and cytoplasmic tRNA$^{Ile}_{UAU}$ signals (n ≥ 50, error bars represent standard deviation, p-value from unpaired t-test with two-tailed distribution shown). (**D**) Northern blot analysis of tRNA$^{Ile}_{UAU}$ (Probe1) as performed in panel A, including an integrated and untagged *dbp5-L12A/R423A* double mutant strain. See *Figure 4—figure supplement 1* for characterization of the *dbp5-L12A/R423A* double mutant strain and tRNA processing defects in Ts alleles of *DBP5, NUP159, GLE1,* and *MEX67*.

The online version of this article includes the following figure supplement(s) for figure 4:

**Figure supplement 1.** Characterization of *dbp5-L12A/R423A* and tRNA processing status in mRNP export mutants.

amino-acid starvation followed by re-feeding through the addition of rich media. Under amino acid starvation, tRNA$^{Tyr}_{GUA}$ rapidly accumulated in the nuclei of both *dbp5-L12A* and *dbp5-R423A* cells in a manner comparable to the wild-type strain, indicating that nuclear tRNA import was not perturbed in these mutants (*Figure 5A*). Following re-feeding, nuclear tRNA levels returned to near pre-stress conditions by 10 minutes in the integrated and untagged versions of the wild-type (15 ± 8) and *dbp5-L12A* (31 ± 5) strains; however, in *dbp5-R423A* the majority of cells at 10 minutes still showed elevated levels of nuclear tRNA (74 ± 9). These observations, plus the genetic interaction with a *msn5Δ*, support the conclusion that nuclear Dbp5 supports the re-export of mature tRNAs following stress. The smaller but significant perturbation in the re-export of tRNAs in *dbp5-L12A* also suggests that efficient nucleocytoplasmic shuttling mediated by the N-terminal NES is important in the context of tRNA re-export.

To further validate these roles for Dbp5 in tRNA biology, in vivo co-immunoprecipitation experiments with protein-A (PrA) tagged Dbp5 followed by RT-qPCR analyses were performed to determine if Dbp5 forms a complex with tRNAs. Using this approach, PrA-Dbp5-RNA interactions were enriched for unspliced and spliced versions of tRNA$^{Ile}_{UAU}$, as compared to an untagged control strain (*Figure 5B* and *Figure 5—figure supplement 1A & B*). PrA-Dbp5$^{R423A}$ interactions appeared reduced with unspliced tRNA$^{Ile}_{UAU}$, but these reductions were not significant at the p<0.05 level (*Figure 5B* and *Figure 5—figure supplement 1A & B*). In contrast, PrA-Dbp5$^{L12A}$ interactions were significantly increased with unspliced tRNA$^{Ile}_{UAU}$ (*Figure 5B* and *Figure 5—figure supplement 1A & B*). These patterns of tRNA$^{Ile}_{UAU}$ binding parallel both the observed export defects in Dbp5$^{R423A}$ and alterations in nuclear transport of both Dbp5$^{L12A}$ and Dbp5$^{R423A}$. In comparison, the highly abundant mitochondrial encoded mRNA *COX1* was not detected by RT-qPCR in any of the co-immunoprecipitation experiments above the background found in negative reverse transcriptase (-RT) controls (*Figure 5—figure supplement 1C*). Overall, the physical interaction of Dbp5 with tRNAs and the observed tRNA export defects in *dbp5-R423A*, which can both be modulated by altering the level of nuclear Dbp5 through use of the L12A mutation, provide strong evidence for Dbp5 functioning within the nucleus in tRNA export.

## Discussion

### Identification of novel functional domains within Dbp5

The work presented here uses a comprehensive mutagenesis strategy to identify separation-of-function alleles in an essential DBP, which contributes to various processes within the gene expression program. Knowledge that can be combined with structural and biochemical data to understand the structure-function relationships between Dbp5 and gene expression. Furthermore, the mutant collection represents a resource that can be employed to identify alleles of value to the study of Dbp5 in particular contexts, and more broadly, to understand DBPs in general. For example, work in this study identified domains important to Dbp5 nuclear transport, including an N-terminal NES sequence. Notably, other DBPs also contain NES sequences at their N-terminus, including multiple *DED1/DDX3* homologs and *DDX28* (*Askjaer et al., 1999*; *Brennan et al., 2018*; *Senissar et al., 2014*; *Valgardsdottir and Prydz, 2003*). Growth data also support the functional importance of critical residues within highly conserved motifs of DBPs required for RNA-binding, ATP-binding, and hydrolysis. In addition, binding sites of Dbp5 specific regulators, Gle1 and Nup159, were identified as lethal when mutated, in agreement with biochemical and structural models describing Dbp5

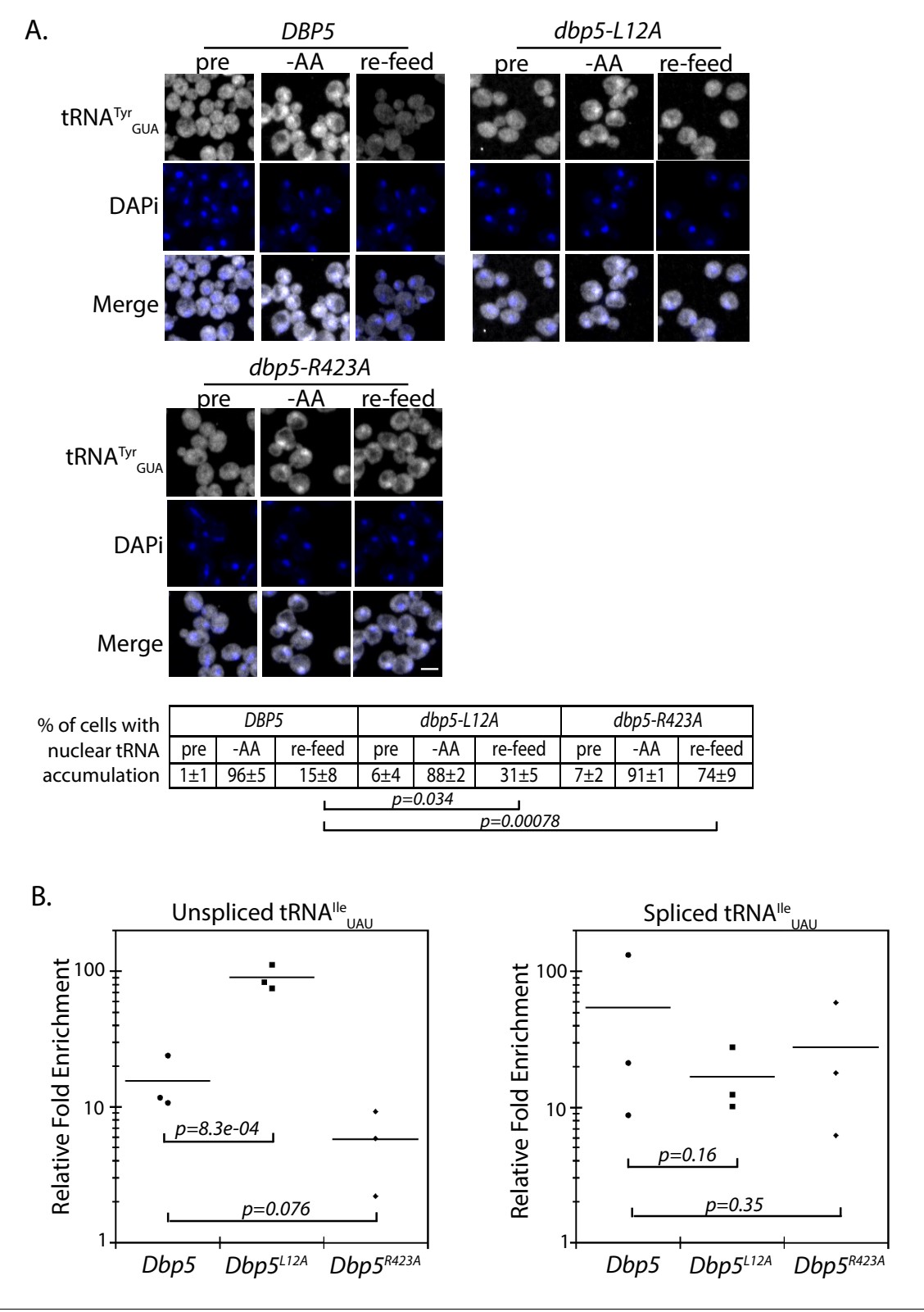

**Figure 5.** Dbp5 supports re-export of mature tRNAs following nutritional stress. (**A**) Localization of tRNA$^{Tyr}$ determined by FISH at 25°C in integrated and untagged *DBP5*, *dbp5-L12A*, and *dbp5-R423A* strains prior to (pre), during a 10 min starvation for amino acids (-AA), and 15 min after reintroduction of amino acids through addition of rich media (re-feed). Cells were probed with a Cy3 end-labeled tRNA$^{Tyr}$ probe (SRIM15, gray) and DAPi (blue). Scale bar = 2 µm. Percent of cells determined to have increased levels of nuclear tRNA (i.e. nuclear signal >than cytoplasmic) under each

*Figure 5 continued on next page*

*Figure 5 continued*

condition are indicated below (three biological replicates, n = 100, error represents standard deviation, p-value from unpaired t-test with two-tailed distribution shown). (B) Graph showing relative fold enrichment (log scale) of associated unspliced or spliced tRNA$^{Ile}_{UAU}$ with PrA-Dbp5 in *DBP5, dbp5-L12A,* and *dbp5-R423A* strains as compared to the untagged control. Data generated by co-immunoprecipitation of PrA-Dbp5 followed by RT-qPCR using formaldehyde cross-linked cells (three biological replicates, p-value from unpaired t-test with two-tailed distribution using delta Ct values). See *Figure 5—figure supplement 1* for silver stained gel showing proteins present in the immunoprecipitation and RT-qPCR products amplified in the analyses of spliced and unspliced tRNA$^{Ile}_{UAU}$, including a control for non-specific binding.

The online version of this article includes the following figure supplement(s) for figure 5:

**Figure supplement 1.** Co-immunoprecipitation analyses show tRNAs co-purify with Dbp5.

regulation (*Alcázar-Román et al., 2006*; *Dossani et al., 2009*; *Hodge et al., 2011*; *Montpetit et al., 2011*; *Noble et al., 2011*; *Weirich et al., 2006*; *Weirich et al., 2004*).

It is important to note that the inclusion of GFP within the mutant collection did result in the modification of mutant phenotypes, which was observed as a Ts and poly(A)-RNA accumulation phenotype at 37°C for both GFP-*dbp5-L12A* and GFP-*dbp5-R423A*. This complicates the interpretation of mutant traits, requiring dissection of the contributions of the mutation vs. GFP to an observed phenotype. For those phenotypes studied in detail here, this was done using an untagged version of the allele integrated at the endogenous *DBP5* locus and expressed from the native promoter. Nonetheless, future screening of this *DBP5* mutant collection against various cellular stresses, or assaying for particular phenotypes, is likely to serve as a powerful means to identify residues that mediate distinct activities of Dbp5 within the gene expression program.

## Nuclear shuttling of Dbp5 and mRNP export

The inclusion of GFP in this collection and subsequent identification of specific mutants that impact nuclear shuttling of Dbp5 was critical to establishing a nuclear function of Dbp5 reported here. This was accomplished through screening the mutant collection in the presence and absence of ethanol stress, which led to the identification of the Xpo1 dependent N-terminal NES sequence in Dbp5 (*Figure 1*) and a stretch of residues in and around motif VI required for efficient nuclear import (*Figure 2*). Interestingly, characterization of NES mutants showed that Dbp5 NES-mediated nuclear export is not essential, and that a large cytoplasmic pool of Dbp5 is not required to support normal cellular functions in the growth conditions tested. This is based on the fact that an integrated and untagged *dbp5-L12A* allele had no growth defect at 25°C or 37°C, no mRNP export defect detectable by FISH, and minimal genetic interactions as measured by SGA. Similarly, *dbp5-R423A* showed no growth defects or detectable poly(A)-RNA accumulation at any temperature. Together these data support the general conclusion that Dbp5 nuclear shuttling is not essential to Dbp5 mRNP export.

Importantly, this work further provides information that can be used to discern current models describing the function of Dbp5 during mRNP export (reviewed in *Heinrich et al., 2017*). In the 'scaffold' model of export, Dbp5 functions as a scaffold required for proper mRNP formation in the nucleus and travels with the mRNP to the cytoplasmic face of an NPC. At the NPC, the Dbp5 interaction with the mRNA would then be altered by the presence of co-regulators, Gle1 and Nup159, resulting in release of the mRNP into the cytoplasm. In this model, efficient shuttling of Dbp5 through the nucleus would be critical for the function of Dbp5 in mRNP export. The alternative 'RNPase' model postulates that Dbp5 first encounters translocating mRNPs at the cytoplasmic face of the NPC and remodels these substrates at this location, an activity that would be independent of a nuclear pool of Dbp5. In this model, a constant pool of Dbp5 at NPCs would be central to mRNP export and altering the residual nuclear or cytoplasmic pools of Dbp5 would have less of an impact on mRNP export. Importantly, the data presented here showed that mRNP export is relatively insensitive to changes in Dbp5 steady-state localization and shuttling, which supports the RNPase model and suggests that the nuclear role of Dbp5 may be more critical for activities relating to other RNA processing and ncRNA export events.

## Dbp5 acts within the nucleus to support tRNA export

Given that Dbp5 has been shown to be required for the export of pre-40s and pre-60s ribosomal subunits and the telomerase RNA *TLC1* (*Neumann et al., 2016*; *Wu et al., 2014*), one hypothesis could be that Dbp5 nuclear shuttling supports these activities. Additionally, export of pre-ribosomal subunits largely occurs through Xpo1 (*Köhler and Hurt, 2007*), and identification of an Xpo1 mediated NES in Dbp5 provides a mode for Dbp5 to act as an export receptor for these cargoes. However, pre-ribosomal subunit export defects in *DBP5* shuttling mutants (*dbp5-L12A* or *dbp5-R423A*) were not observed under the conditions tested. This may be due in part to multiple redundant pathways that contribute to pre-ribosomal subunit export, which includes Arx1, and the essential non-karyopherin transport receptor required for mRNP export, Mex67 (TAP or NXF1) (*Bradatsch et al., 2007*; *Faza et al., 2012*; *Hung et al., 2008*; *Yao et al., 2007*). The lack of a discernible phenotype is also in agreement with work from *Neumann et al. (2016)*, which suggests that Dbp5 activity in relation to rRNA export is occurring at the cytoplasmic face of NPCs, similar to mRNP export. Given that both Dbp5$^{L12A}$ and Dbp5$^{R423A}$ still access the nuclear envelope, a block in rRNA may not be expected.

Instead, this work identified a requirement for nuclear Dbp5, and Dbp5 shuttling, in tRNA export (*Figures 3–5*). Specifically, analyses of wild-type and mutant strains provided evidence for a direct role for nuclear Dbp5 in tRNA export, including a physical interaction between Dbp5 and tRNA transcripts. tRNA export defects were most apparent in an integrated *dbp5-R423A* strain after a temperature shift to 37°C or following addition of nutrients (e.g. amino acids) to starved cells. Re-introduction of nutrients is a condition that induces re-export of a large nuclear pool of mature tRNAs that accumulate during nutrient starvation (*Whitney et al., 2007*), which was delayed in *dbp5-R423A*. Notably, a synthetic genetic interaction between *dbp5-R423A* and a *msn5Δ* was observed by SGA, with Msn5p also known to promote tRNA re-export following nutrient stress (*Huang and Hopper, 2015*; *Murthi et al., 2010*). It is expected that under conditions of high temperature growth or nutrient stress tRNA export defects are most apparent due to the demands put on the tRNA export machinery, which may result from altered tRNA export due to regulation in response to stress and/or lowered efficiencies of other parallel tRNA export pathways under these conditions.

In the case of Dbp5$^{R423A}$, one potential mechanism by which this mutation could impact tRNA export would be through reducing access of Dbp5 to the nuclear compartment. For example, lowered import rates in Dbp5$^{R423A}$ coupled with efficient nuclear export may result in a reduced steady-state level of nuclear Dbp5$^{R423A}$ that is not sufficient for tRNA export. However, while binding interactions with unspliced tRNAs appear to be lowered in Dbp5$^{R423A}$, interactions were still detected and were not significantly different from wild-type at the $p<0.05$ level. As such, altered nuclear access of Dbp5 may not be the only way in which this mutation alters tRNA export, with a second possible means being the altered ability of RNA to stimulate Dbp5$^{R423A}$ ATPase activity. In this context, while binding events occur between tRNA and Dbp5$^{R423A}$, as detected in pull-down assays, these interactions may not be coupled with a productive ATPase cycle. This would potentially delay tRNA export due to the need for multiple rounds of Dbp5 binding to a tRNA before ATPase hydrolysis successful occurs and the work required for export is performed. However, it is most likely that the combined lack of nuclear access and altered ATPase activities are the cause of the observed defects in tRNA export. This is based on the fact that the ATPase defect in Dbp5$^{R423A}$ is not associated with an mRNP export defect, suggesting ATPase rates are sufficient to meet the needs of mRNA export under conditions where access to the cytoplasmic face of NPCs is unchanged. However, other possibilities exist, including the possibility that tRNA export requires a higher overall level of Dbp5 ATPase activity or the activity required for tRNA export is fundamentally different than mRNP export, either of which are activities unmet by Dbp5$^{R423A}$. Future studies will pursue mechanisms by which Dbp5 mediates tRNA export, especially in response to changes in nutrient availability in conjunction with Msn5p, and how these functions of Dbp5 are specifically impacted in *dbp5-R423A*.

## Dbp5, Nup159, Gle1, and Mex67 in tRNA export and beyond

Previous work has also identified a role for the mRNP export factor Mex67 and the Dbp5 regulator Nup159 in the export of tRNA substrates (*Chatterjee et al., 2017*; *Wu et al., 2015*). Here, the role

of another Dbp5 regulator, Gle1, in tRNA export was identified. The fact that these four proteins, which function together to support mRNP export (*Folkmann et al., 2011*; *Heinrich et al., 2017*; *Stewart, 2007*), also show tRNA export defects raises the possibility that this machinery works together to facilitate the export of multiple transcript classes. Importantly, while *nup159-1*, *gle1-4*, and *dbp5-1* Ts alleles have strong mRNP export defects at 37°C, *dbp5-R423A* does not. In addition, tRNA export defects are apparent in *gle1-4*, *nup159-1*, and *dbp5-1* strains at 25°C in the absence of a block to mRNP export. This lends support to the postulate that Dbp5, and both regulators Gle1 and Nup159, have roles in tRNA export that are independent of a block in mRNP export.

Given this work, many questions emerge surrounding the function(s) of Dbp5, and how this activity is modulated in vivo, to support mRNA, tRNA, and rRNA export. For example, does Dbp5 generally function to remodel all RNP complexes exiting NPCs or does Dbp5 function in a distinct manner on mRNA, tRNA, and rRNA substrates (e.g. RNPase vs. scaffold models)? For mRNAs, data presented here supports the conclusion that Dbp5 functions at NPCs during mRNP export, in line with an RNPase model (*Figure 6*). In contrast, the requirement for nuclear Dbp5 to support tRNA export leads to two broad non-mutually exclusive models, which include Dbp5 acting solely in the nucleus as an RNPase or Dbp5 acting as a scaffold and traveling with tRNA to the cytoplasm (*Figure 6*). Notably, Mex67 is known to support the export of mRNA, tRNA and rRNA (*Bradatsch et al., 2007*; *Hurt et al., 2000*; *Segref et al., 1997*; *Wu et al., 2015*), and biochemical evidence suggests that Dbp5 targets Mex67 on mRNPs (*Lund and Guthrie, 2005*). As such, does Dbp5 target Mex67 bound to all these RNA substrates? Finally, an important question to address is the role of Dbp5 during nutrient stress and recovery? Do these activities of Dbp5 during a stress response extend to other RNA substrates to more globally regulate the gene expression program? Further work is required to address all of these questions and it is expected that the mutants characterized here, and potentially others within the collection, will serve as a rich resource to aid in this future work.

## Materials and methods

### Strains, plasmids, and oligos
For a list of DBP5 mutagenesis primers, yeast strains, plasmids, qPCR primers, FISH probes, and Northern probes see *Supplementary file 1* - Tables 3-6.

### Scanning mutagenesis
A plasmid (pBM464) containing DBP5 ±500 bp of flanking sequence was generated with an N-terminal GFP sequence fused to the DBP5 coding region as a template for mutagenesis. Primers for alanine scanning mutagenesis were designed using the AAscan program against all residues (*Sun et al., 2013*). PCR was carried out using Q5 high fidelity enzyme (NEB) in 12.5 µl reactions, followed by DpnI (NEB) digestion for 2 hr. Digested PCR product (5 µl) was transformed into DH5α competent *E. coli* cells and plated onto LB ampicillin in a 12 well plate format. Mini preps were performed in a 96 well format (BioBasic Inc) on selected single colonies to purify mutagenized plasmids. The presence of each mutation was verified by sequencing (Genome Quebec).

### Yeast mutant library generation
Sequence verified plasmids were transformed into yeast in a 96-well plate format using a *dbp5Δ* strain carrying DBP5 on a URA3 marked CEN plasmid (BMY015). The BMY015 strain was grown to an O.D.$_{600}$ of 1.0 in 1.5 L of selective media, centrifuged at 3000xg for 5 min, washed once in water, and then in 50 ml of 100 mM lithium acetate (LiAc). The culture was re-suspended in 30 ml of 100 mM LiAc and 50 µl of culture was aliquoted into each well of a 96-well plate. Plates were centrifuged at 3000xg for 5 min and supernatant was removed. Cells were re-suspended in a transformation mix (80 µl of 50% PEG, 12 µl of 1.0 M LiAc, 3.3 µl of 10 mg/ml salmon sperm single stranded DNA, and 22.5 µl of water). Mutated plasmid DNA (250 ng) was added to each well and mixed. Plates were incubated at 30°C for 45 min and then at 42°C for 30 min. Plates were centrifuged again and the supernatant was removed and 150 µl of synthetic complete (SC)-LEU/-URA media was added. Cultures were grown for 3 days at 25°C and then 5 µl of culture was diluted into 200 µl of fresh media and grown for another 2 days. Transformants were then pinned for two rounds onto SC/-LEU/-URA selective agar plates and grown at 25°C. To remove the wild-type plasmid, strains were also pinned

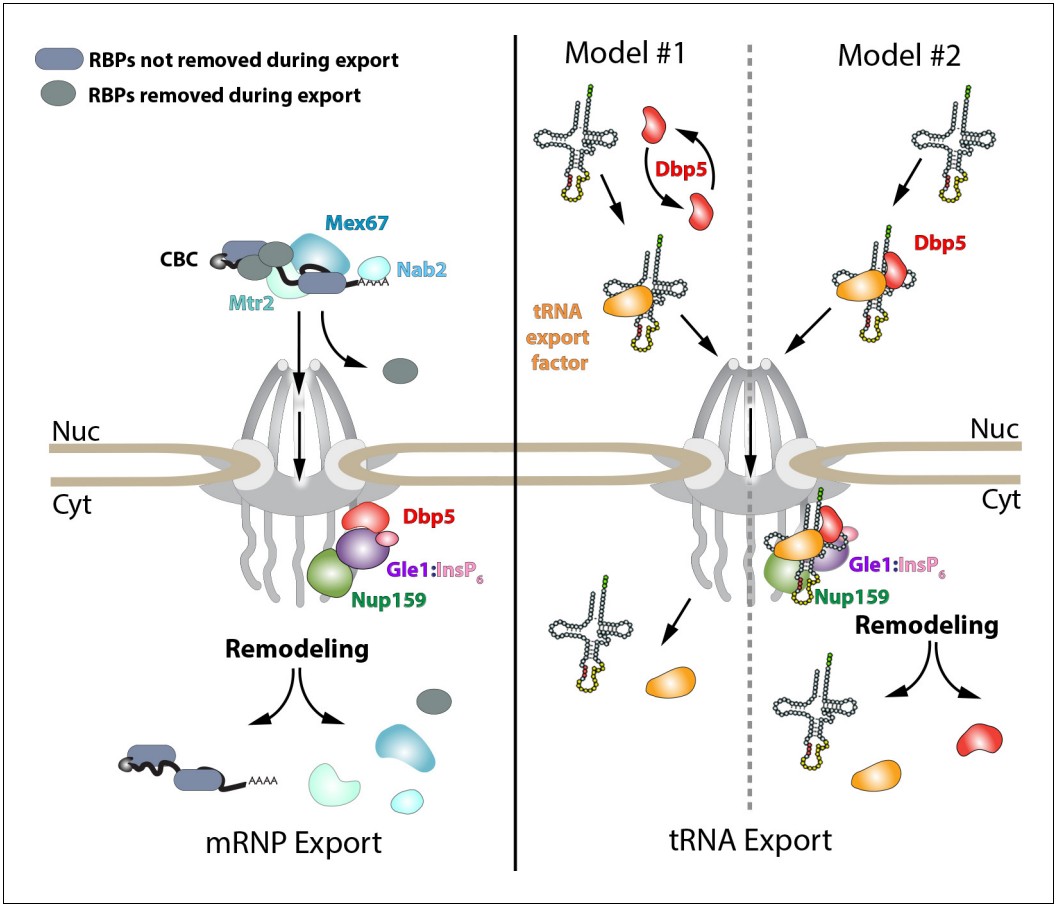

**Figure 6.** Models of Dbp5 function in mRNP and tRNA export. Data from this work suggest that Dbp5 shuttling is not critical to mRNP export, supporting a model in which Dbp5 acts at NPCs to facilitate export. In such a model (schematic on left), mRNPs assembled in the nucleus would be expected to include the cap binding complex (CBC), export factors Mex67/Mtr2, Nab2, and various other RBPs. Following transport out of the nucleus, the activity of Dbp5 at the cytoplasmic face of the NPC in the context of Gle1:InsP$_6$ and Nup159 would promote remodeling of RNA-RBP interactions to enforce directional transport (e.g. RNPase model). For tRNAs, a nuclear pool of Dbp5 is involved in export suggesting two broad models of Dbp5 function (schematic on right). In model #1, Dbp5 would act solely within the nucleus to support tRNA processing and export by facilitating events, potentially as an RNPase, that ultimately lead to a tRNA being exported with a tRNA export factor (e.g. Los1 or Mex67). In model #2, Dbp5 engages a tRNA in the nucleus and travels with tRNAs from the nucleus to the cytoplasm in complex with the tRNA and tRNA export factor (e.g. scaffold model), and upon entering the cytoplasm, these interactions would be remodeled in the context of the NPC and Dbp5 regulators. The second model provides a rationale for tRNA export defects in mutants of Dbp5 co-regulators (Nup159 and Gle1), whereas in model #1 the impact of mutations in Dbp5 co-regulators would act through an indirect and/or independent mechanism.

on to 5-FOA for two rounds of selection before finally pinning onto SC/-LEU plates. Strains that did not grow following plating on 5-FOA were considered to contain lethal mutations. To determine which mutant strains were temperature sensitive, the mutant array was plated on SC –LEU plate, grown at 37˚C for 3 days, and then repined on SC –LEU and grown at 37˚C for 3 days. Strains that did not grow when compared to a 25˚C control plate were noted to be temperature sensitive and were subsequently verified by additional spot plating assays at 37˚C.

## Live-cell fluorescence microscopy and image analysis

Live cell imaging was performed using a widefield or confocal configuration on an Andor Dragonfly microscope equipped with an EMCCD camera driven by Fusion software (Andor) with a 60x oil immersion objective (Olympus, numerical aperture [NA] 1.4). For imaging of the mutant collection,

strains were grown overnight in 96 well plates at 25°C to mid-log phase in SC media and then placed in 384-well glass bottomed plates (VWR) treated with Concanavalin A. To determine changes in localization as a result of ethanol stress, media containing 24% ethanol was added to each plate well to achieve a final concentration of 12% ethanol, followed by imaging 15 min after ethanol addition. Anchor away experiments were similarly performed in glass bottom plates using 1 μg/ml rapamycin, pre-treated with or without 100 ng/ml leptomycin B, followed by imaging for 20 min at 1 min intervals. To quantify the nuclear/cytoplasmic fluorescence intensity of GFP-NLS/NES reporters, the mean integrated fluorescence intensity was measured for a 0.5 μm x 0.5 μm region in the nucleus or cytoplasm. Signal intensity was normalized to a background fluorescent intensity value adjacent to each cell and used to calculate the nuclear/cytoplasmic ratios. For each condition, measurements for 100 cells were obtained and the average ratios were plotted in a bar graph with the variability between cells expressed as standard deviation. Image analyses were performed in FIJI, including maximum z-projections, image cropping, and brightness/contrast adjustments (*Schindelin et al.,* *2012*).

## Silver staining

SDS-PAGE was carried out as described above. Gels were fixed with 50% ethanol 2x for 15 min followed by 20 min incubation in DTT (5 μg/ml) and a 20 min incubation in 0.1% AgNO3. Finally, the gel was developed in developing solution (3% Na2CO3 (w/v), 2% formaldehyde) and stopped by incubation in 1% Acetic Acid.

## Fluorescence in situ hybridization (FISH)

FISH was carried out as described with modifications (*Chen et al., 2018*). Briefly, strains were grown overnight to mid-log phase at 25°C and in some cases shifted to 37°C for indicated time points. Samples were fixed with formaldehyde overnight at a final concentration of 3.7%. Samples were then washed three times with ice cold Buffer B (1.2M Sorbitol, 0.1M Potassium Phosphate, 0.5 mM MgCl$_2$). To spheroplast cells, 2.5 μL 20 mg/mL zymolyase T20 and 5 μL of 200 mM vanadyl ribonucleoside complex (VRC) were added to cells re-suspended in 425 μL of Buffer B for 30 min at 30°C. Samples were washed once with Buffer B and then re-suspended in 70% ethanol and incubated for 4 hr at room temperature. After washing samples with Buffer B, hybridization was carried out in 50 ul of buffer containing 1X saline sodium citrate (SSC), 0.34 mg/ml *E. coli* tRNA, 20% formamide, 0.2 mg/ml BSA, 11% dextran sulphate, 4 mM VRC, and oligo-dT or gene specific probes as indicated at 37°C for 16 hr. A final concentration of 0.4 mM was used for the fluorescein isothiocyanate (FITC) labeled oligo dT probe (Qiagen). Samples were washed two times with 1X SSC and 15% formamide followed by application onto 8-well slides (Fisher Scientific) treated with Poly-L-Lysine (SIGMA). After washing twice with 1x phosphate buffered saline (PBS), mounting medium with DAPI was applied to each sample and a coverslip was affixed. For tRNA specific FISH was carried out as previously described by *Chatterjee et al. (2017)* with some modifications. Briefly, samples were grown to early log phase (~O.D.$_{600}$=0.3) in YPD at 25°C (or in some cases in YPD at 37°C or in SC media lacking leucine, histidine, uracil, methionine and tryptophan) followed by formaldehyde fixation at a final concentration of 3.7% for 15 min. Samples were then re-suspended in 5 ml of fresh Buffer A (2% paraformaldehyde, 0.1M Potassium Phosphate, 0.5 mM MgCl$_2$) for 3 hr at room temperature. Cells were then washed two times with Buffer B. To spheroplast cells, 8 μL of 20 mg/mL zymolyase T20 and 1 μL of β-mercaptoethanol was added to cells re-suspended in 500 μL of Buffer B for 30 min at 37°C. Cells were then washed one time with Buffer B and adhered to eight well slides coated with Poly-L-lysine. Slides were treated with 70%, 90%, and 100% ethanol for 5 min. Cells were then incubated in a pre-hybridization buffer (10% dextran sulfate, 0.2% BSA (acetylated), 2 × SSC (1 × SSC is 0.15 M NaCl and 0.015 M Na-citrate), 1X Denhardt's solution, 250 μg *Escherichia coli* tRNA/ml) for 2 hr at 37°C. For hybridization, 1 pmol/ul of Cy3-dUTP end labeled tRNA probes were added to fresh hybridization buffer and incubated overnight at 37°C. All tRNA FISH probes were 3'end labeled using an oligonucleotide end labeling kit (Roche) and Cy3-dUTP (Enzo Life Sciences). Cells were washed one time in 2xSSC at 50°C, and then two subsequent washes for 10 min at room temperature each in 1X SSC, 4X SSC, 4X SSC +1% Triton X-100. Slides were then washed in 4X SSC again briefly, and once in 1X PBS and then mounted with DAPi and sealed. Imaging was performed using a microscope (Dragonfly; Andor) equipped with an EMCCD camera driven by Fusion (Andor) using a

60x × 1.4 NA oil objective (Olympus). Image analysis was performed in FIJI, including maximum z-projections, image cropping, and brightness/contrast adjustments (*Schindelin et al., 2012*).

## Immunofluorescence (IF)

Strains were grown overnight to mid-log phase at 25°C and were fixed with formaldehyde at a final concentration of 3.7% for 15 min. Samples were then washed two times with Buffer A (10.1M Potassium Phosphate, 0.5 mM $MgCl_2$) and re-suspended in 1 ml of Buffer B (1.2M Sorbitol, 0.1M Potassium Phosphate, 0.5 mM $MgCl_2$). To spheroplast cells, 2.5 μL 20 mg/mL zymolyase T20 and 1 μL of β-mercaptoethanol were added to cells for 30 min at 37°C. Samples were washed once with Buffer B and then adhered to eight well slides coated with Poly-L-lysine. Slides were treated ice-cold methanol for 6 min and then acetone for 30 s. Samples were blocked with 1X phosphate buffered saline with 5 mg/ml bovine serum albumin (PBS-BSA) for 1 hr at room temperature. Samples were incubated with a mouse monoclonal anti-DBP5 primary antibody overnight at room temperature in PBS-BSA with 0.05% Tween-20. Slides were then washed five times with PBS-BSA-Tween for 5 min each and then incubated with a Goat anti-mouse DyLight 650 (Thermo Fisher) secondary antibody in PBS-BSA-Tween. Slides were then washed five times with PBS-BSA-Tween for 5 min each, and twice in 1X PBS and then mounted with DAPi and sealed. Imaging was performed using a microscope (Dragonfly; Andor) equipped with an EMCCD camera driven by Fusion (Andor) using a 60x × 1.4 NA oil objective (Olympus). Image analysis was performed in FIJI, including maximum z-projections, image cropping, and brightness/contrast adjustments (*Schindelin et al., 2012*).

## Synthetic gene array screen

SGA screening was carried out as previously described (*Young et al., 2010*). Query strains were mated to the yeast deletion mutant array (DMA) at a density of 1536 spots per plate using a Singer RoToR HDA pinning robot. Following selection, diploids were sporulated and then germinated on SD-His/Arg/Lys media supplemented with 100 mg/L canavanine, 100 mg/L thialysine, 200 mg/L G418 sulphate (HRK media) to select for MATa haploid cells. Control data sets was generated by pinning the array to HRK media + 1 g/L 5-fluorooritic acid, followed by a further pinning to HRK media. The experimental data sets were generated by two rounds of pinning to HRK media without uracil to select for strains carrying *dbp5* alleles. Images of plates were captured on a flatbed scanner and analyzed using *Balony* software (*Young and Loewen, 2013*).

## Protein purification and ATPase assay

Protein purification and in vitro ATPase assays carried out as previously described using full-length Dbp5 or Dbp5-R423A (*Montpetit et al., 2011*).

## RNA extraction

tRNAs were extracted from yeast cultures as described previously by *Hopper et al. (1980)*. Briefly, strains were grown overnight to early log phase and harvested. Cell pellets were re-suspended in equal volumes of ice-cold TSE buffer (0.01M Tris pH7.5, 0.01M EDTA, 0.1M sodium chloride) and TSE saturated phenol. The mixture was incubated for 20 min at 55°C and vortexed every 3 min. Phases were separated by centrifugation at 20 000 g for 10 min and re-extracted with phenol. RNA was precipitated overnight in ethanol at −80°C.

## Northern blotting

Northern blotting was carried out as previously described by *Chatterjee et al. (2017)*. 2.5 micrograms of total RNA was separated by electrophoresis on 10% TBE-Urea gels and transferred onto a Hybond $N^+$ membrane (Amersham). Membranes were cross-linked at 2400 $J/m^2$ (UV Crosslinker, VWR). tRNAs were detected using with digoxigenin-labeled (DIG) probes as described previously (*Wu et al., 2015*). Gels were stained with 1 μg/ml ethidium bromide to detect 5.8S and 5S rRNAs as loading controls. Mean integrated intensities of the bands detected were measured using FIJI (*Schindelin et al., 2012*), and normalized to a background signal for each lane. These values were then used to calculate the ratio between the intron containing tRNA species and the precursor tRNA species and normalized to the value calculated for the wild-type control sample.

## RNA immuno-precipitation (RIP)

RIP experiments were performed with Protein-A tagged Dbp5 strains, in parallel with an untagged control strain to assess background binding of non-specific RNAs following IP. Strains were grown overnight to mid-log phase at 25°, formaldehyde was added to a final concentration of 0.3%, and incubated for 30 min. Cross-linking was quenched by addition of glycine to a final concentration of 60 mM and incubated for 10 min. Cells were harvested and flash frozen in liquid nitrogen. Pellets were re-suspended in 1.5 ml TN150 lysis buffer (50 mM Tris-HCl pH 7.8, 150 mM NaCl, 0.1% IGE-PAL, 5 mM beta-mercaptoethanol), Protease Inhibitor Cocktail was added to 1X concentration and lysis was performed by vortexing with 1 ml Zirkonia beads (0.5 mm) 5 times for 1 min with 1 min incubation on ice between each cycle. 5 ml TN150 was added and lysate was cleared by centrifugation (20 min at 4000xg; then 20 min at 20000xg; 4°C). Lysates were diluted to 10 ml with TN150 and incubated with IgG-conjugated magnetic dynabeads at 4°C 1 hr with rotation. Immuno-precipitate was washed once for 5 min with 1 ml TN150, followed by 5 min 1 ml TN1000 (50 mM Tris-HCL pH7.8, 1M NaCl, 0.1% IGEPAL, 5 mM beta-mercaptoethanol), and once more with 1 ml TN150 for 5 min at 4°C with rotating. Immuno-precipitate was re-suspended in 100 µl TurboDNase reaction (1X TurboDNase Buffer, 2U Turbo DNase) and incubated on a thermomixer at 37°C for 30 min at 1200 RPM. Magnetic beads were washed 3 times with 1 ml TN150 and 50 µl of beads were retained for western blot analysis. Remaining beads were aspirated of TN150 and re-suspended in 400 µl Proteinase K elution mix spiked with exogenous Luciferase RNA (50 mM Tris HCL pH7.8, 50 mM NaCl, 1 mM EDTA, 0.5% SDS, 1 ng Promega Luciferase Spike-In RNA, 100 µg proteinase K). Elution was carried out at 50°C 1200 rpm for 2 hr on thermomixer. Magnetic beads were incubated at 65°C for 1 hr to allow cross-link reversal by heat. RNA was extracted from IP with 400 µl Sigma 5:1 phenol:chloroform pH 4.3–4.7 followed by back extraction with chloroform:isoamylalcohol and ethanol precipitation for 1 hr at −80°C using linear acrylamide as a carrier. Half volume of RIP RNA was reverse transcribed with Superscript III using random priming, while the other half was retained for -RT reaction to assess genomic contamination. Reverse transcription was carried out as per manufacturer's instructions. Resulting cDNA from RIPs was used in standard PCR followed by 2% gel electrophoresis in addition to being used as template in qPCR to quantify fold enrichment of target RNAs in RIP.

## RT-qPCR analysis of RIPs

RT-qPCR experiments were carried out using Power SYBR (Applied Biosystems) on an Applied Biosystems instrument. Target RNA abundance in each RIP was normalized to the abundance of an exogenous spike-in Luciferase RNA control to correct for differences in sample preparation (dCt). ddCt values were then calculated as the difference in relative abundance of the target RNA isolated via immunoprecipitation using a Protein-A tagged Dbp5 strain vs. an untagged strain (non-specific binding control). Relative fold-enrichment above the untagged background control was calculated using the Livak Method (*Livak and Schmittgen, 2001*). Standard curves were generated to test PCR efficiencies of all primer sets used in this study. Averages, SD, and p-values were calculated on dCt values across three independent RIP experiments.

## Acknowledgements

We would like to acknowledge Drs. Gary Eitzen (University of Alberta), Rick Wozniak (University of Alberta), and Anita Hopper (Ohio State University) for reagents and protocols used in this work. We also thank Kelly Tedrick, Brenda Lam, and all past and current members of the Montpetit laboratory for their support of this work. AL was supported by an Alexander Graham Bell Canada Graduate Scholarship, an Alberta Innovates Technology Futures Graduate Scholarship, and a Canada Graduate Scholarships – Michael Smith Foreign Study Supplement. AANR was funded by the predoctoral Training Program in Molecular and Cellular Biology at UC Davis that is supported by an NIH T32 training grant (GM007377). TR was supported by the Gordon and Betty Moore Foundation's Data-Driven Discovery Initiative through Grant GBMF4551, Harry Baccigaluppi Fellowship, the Horace O Lanza Scholarship, Louis R Gomberg Fellowship, the Margrit Mondavi Fellowship, the Haskell F Norman Wine and Food Fellowship, the Chaîne des Rôtisseurs Scholarship, and the Carpenter Memorial Fellowship. Research reported in this publication was supported by the National Institute of General Medical Sciences of the National Institutes of Health under Award Number R01GM124120 (BM) and

by Canadian Institutes of Health Research grants MOP 130231 (BM) and 79497 (CJRL). The content is solely the responsibility of the authors and does not necessarily represent the official views of the National Institutes of Health or other funding agencies.

## Additional information

### Funding

| Funder | Grant reference number | Author |
|---|---|---|
| National Institute of General Medical Sciences | R01GM124120 | Ben Montpetit |
| Canadian Institutes of Health Research | 130231 | Ben Montpetit |
| Canadian Institutes of Health Research | 79497 | Chris JR Loewen |

The funders had no role in study design, data collection and interpretation, or the decision to submit the work for publication.

### Author contributions

Azra Lari, Conceptualization, Data curation, Formal analysis, Investigation, Methodology, Writing—original draft, Writing—review and editing; Arvind Arul Nambi Rajan, Rima Sandhu, Formal analysis, Investigation, Methodology, Writing—review and editing; Taylor Reiter, Formal analysis, Methodology, Writing—review and editing; Rachel Montpetit, Investigation, Methodology, Writing—review and editing; Barry P Young, Formal analysis, Investigation, Methodology; Chris JR Loewen, Supervision, Project administration; Ben Montpetit, Conceptualization, Funding acquisition, Investigation, Methodology, Writing—original draft, Project administration, Writing—review and editing

### Author ORCIDs

Azra Lari (iD) https://orcid.org/0000-0002-9649-8231
Taylor Reiter (iD) http://orcid.org/0000-0002-7388-421X
Ben Montpetit (iD) https://orcid.org/0000-0002-8317-983X

### Decision letter and Author response

Decision letter https://doi.org/10.7554/eLife.48410.sa1
Author response https://doi.org/10.7554/eLife.48410.sa2

## Additional files

### Supplementary files

• Supplementary file 1. File contains the following tables referenced in the main text. Table 1: List of lethal point mutations in Dbp5, Table 2: results of the SGA analysis of *dbp5-L12A* and *dbp5-R423A*, Table 3: *DBP5* mutagenesis oligo sequences, Table 4: Yeast Strains, Table 5: Plasmids, and Table 6: sequences of qPCR primers, FISH probes, and Northern probes.

• Transparent reporting form

### Data availability

All data generated or analysed during this study are included in the manuscript and supporting files.

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
