## [Decision Letter]

Thank you for submitting your article "A nuclear role for the DEAD-box protein Dbp5 in tRNA export" for consideration by *eLife*. Your article has been reviewed by Kevin Struhl as the Senior Editor, a Reviewing Editor, and two reviewers, a Reviewing Editor, and two reviewers. The reviewers have opted to remain anonymous.

The reviewers have discussed the reviews with one another and the Reviewing Editor has drafted this decision to help you prepare a revised submission.

The authors conduct a comprehensive analysis of the roles of the budding yeast DEAD box protein, Dbp5, in RNA nuclear export, RNA response to environmental stress, and protein synthesis. The studies are based upon an alanine scanning mutant collection they created and their extensive analyses including: growth phenotypes, live cell imaging of the localization mutant tagged Dbp5 proteins, protein localization motifs via reporter constructs, Sup45 co-IP, subcellular protein tethering, RNA FISH localization of poly A mRNA and tRNA, tRNA co-purification with Dbp5; RNA seq changes in response to stress, and SGA for two dbp5 alleles!

Due to the massive data sets and analyses the manuscript is quite exhausting to read and the reviewers fear that it will be to disperse and out of reach for non-experts. Still, the authors report results of scientific interest and validity. Hence, to make the paper scientifically appropriate for *eLife* a significant re-writing is suggested. The main point of the manuscript, a role of Dbp5 in tRNA metabolism, is appropriately supported. Moreover, the different roles of Dbp5 in mRNA nuclear export vs. tRNA nuclear export are interesting, including: the different motifs important for both, the data indicating that nucleus-cytoplasm Dbp5 shuttling seems not to be important for mRNA nuclear export (supporting the model of Dbp5 functioning to release mRNA on the cytoplasmic surface), whereas it appears to be important for tRNA nuclear export (support a model of Dbp5 assembling tRNA in an export complex).

In contrast, the reviewers believe, that the sections describing the RNA-seq analyses, the translation inhibitor studies and MMS screen do not make a coherent scientific point, but could potentially be assembled into separate manuscripts.

1) Tailoring of the manuscript throughout would really help its impact. Models/cartoons – diagrams often convey messages more effectively than text. For example, a cartoon indicating how Dbp5 may function differently in mRNA vs. tRNA nuclear export could help readers understand the complexity of Dbp5's biological roles and the roles of different motifs in the various functions.

2) Figure labeling – it would be easier to access the data if the labeling of figures were improved. For example: Figure 2—figure supplement 1B and C appear similar, but B is FISH and C is protein imaging; so, putting this information on the figure in addition to the legend would be helpful to the readers. Another example for Figure 5; it would be helpful to label A. tRNAIle, B. tRNATyr, C. FISH tRNATyr, D. tRNAIle. Also, it is not clear in the figure or the text that A and D were conducted in Dbp5-tagged vs. untagged strains.

3) tRNA experiments: Figure 5C and D; why is the FISH conducted for tRNATyr whereas the northern for tRNAIle? Figure 6, even though L12A doesn't have a very remarkable affect upon tRNA nuclear accumulation, its response to -AA likely should be included in Figure 6. tRNA RIP analyses should include an internal nonspecific RNA, such as a mitochondrial tRNA, to assess pull-down specificity. One would have expected that L12A and R423A would have shown different levels of tRNA pulldown given their different roles as analyzed by FISH; how do the authors think about this.

4) Subsection “Gene expression in *dbp5-L12A* and *dbp5-R423A* at steady-state and in response to MMS”: no data are shown for the claim that steady state RNA levels were similar for wt, L12A and R423A mutants.

5) Figure 4 – it is nearly impossible to read the identities of the genes through the magenta and purple colors.

6) Figure 4A,B: differences in RNA levels in the mutants upon MMS treatment would be better shown as correlation plots. The data are hard to parse in their current form.

7) Subsection “Gene expression in *dbp5-L12A* and *dbp5-R423A* at steady-state and in response to MMS”: The FDR values for the highlighted GO terms are large (i.e. barely significant), raising the question whether the reported enrichments are biological meaningful. The authors make the point that L12A and R423A cause opposite changes in gene expression in the GO categories. Are the same genes affected? If so, this should be shown. If not, the FDR values for enrichment would probably render this effect statistically insignificant. More depth is needed in this section.

8) Subsection “*dbp5-L12A* and *dbp5-R423A* genetic interaction profiles”: The referenced Figure 4 – figure supplement 1 seems to be missing.

[Editors' note: further revisions were requested prior to acceptance, as described below.]

Thank you for resubmitting your work entitled "A nuclear role for the DEAD-box protein Dbp5 in tRNA export" for further consideration at *eLife*. Your revised article has been favorably evaluated by Kevin Struhl (Senior Editor), a Reviewing Editor, and two reviewers.

The manuscript has been improved and acceptance is recommended in principle, but there is one remaining issue that needs to be addressed before acceptance, The new Figure 6 does not provide identification for all the proteins, either on the figure or in the legend.

---

## [Author Response]

The reviewers have discussed the reviews with one another and the Reviewing Editor has drafted this decision to help you prepare a revised submission.The authors conduct a comprehensive analysis of the roles of the budding yeast DEAD box protein, Dbp5, in RNA nuclear export, RNA response to environmental stress, and protein synthesis. The studies are based upon an alanine scanning mutant collection they created and their extensive analyses including: growth phenotypes, live cell imaging of the localization mutant tagged Dbp5 proteins, protein localization motifs via reporter constructs, co-IPs, subcellular protein tethering, RNA FISH localization of poly A mRNA and tRNA, tRNA co-purification with Dbp5; RNA seq changes in response to stress, and SGA for two dbp5 alleles!

We wish to thank the reviewers and editors for their constructive and helpful comments. Please find below our response to each.

Due to the massive data sets and analyses the manuscript is quite exhausting to read and the reviewers fear that it will be to disperse and out of reach for non-experts. Still, the authors report results of scientific interest and validity. Hence, to make the paper scientifically appropriate for eLife a significant re-writing is suggested. The main point of the manuscript, a role of Dbp5 in tRNA metabolism, is appropriately supported. Moreover, the different roles of Dbp5 in mRNA nuclear export vs. tRNA nuclear export are interesting, including: the different motifs important for both, the data indicating that nucleus-cytoplasm Dbp5 shuttling seems not to be important for mRNA nuclear export (supporting the model of Dbp5 functioning to release mRNA on the cytoplasmic surface), whereas it appears to be important for tRNA nuclear export (support a model of Dbp5 assembling tRNA in an export complex).In contrast, the reviewers believe, that the sections describing the RNA-seq analyses, the translation inhibitor studies and MMS screen do not make a coherent scientific point, but could potentially be assembled into separate manuscripts.

The manuscript has been rewritten to focus on Dbp5 nuclear transport and the role of Dbp5 in tRNA metabolism. This includes removal of the phenotypic screens and RNA-seq data as suggested. We believe this has improved overall of clarity and more effectively conveys the impact of the work. Again, we thank the reviewers for this suggestion.

1) Tailoring of the manuscript throughout would really help its impact. Models/cartoons – diagrams often convey messages more effectively than text. For example, a cartoon indicating how Dbp5 may function differently in mRNA vs. tRNA nuclear export could help readers understand the complexity of Dbp5's biological roles and the roles of different motifs in the various functions.

As noted above, rewriting has aided with both message and impact. We have also added a model figure to help convey potential ways Dbp5 is functioning in tRNA vs. mRNA export (see Figure 6).

2) Figure labeling – it would be easier to access the data if the labeling of figures were improved. For example: Figure 2—figure supplement 1B and C appear similar, but B is FISH and C is protein imaging; so, putting this info. on the figure in addition to the legend would be helpful to the readers. Another example for Figure 5; it would be helpful to label A. tRNAIle, B. tRNATyr, C. FISH tRNATyr, D. tRNAIle. Also, it is not clear in the figure or the text that A and D were conducted in Dbp5-tagged vs. untagged strains.

We have added labels as requested, plus reviewed all figures and legends to improve labeling and clarity.

3) tRNA experiments: Figure 5C and D; why is the FISH conducted for tRNATyr whereas the northern for tRNAIle?

Figure 5C/D, now Figure 4C/D, were both performed with tRNAIle, which was not apparent based on figure labeling. Related to comment #2 above, we have now added labels to resolve this issue.

Figure 6, even though L12A doesn't have a very remarkable affect upon tRNA nuclear accumulation, its response to -AA likely should be included in Figure 6.

We have added L12A data as requested. Please see current Figure 5 (previously Figure 6).

tRNA RIP analyses should include an internal nonspecific RNA, such as a mitochondrial tRNA, to assess pull-down specificity.

We have now assessed pulldown specificity by looking for the mitochondrial COX1 mRNA in cDNA libraries generated from all RIP experiments used to generate the data presented in Figure 5B. The COX1 transcript was not detected in any of the Dbp5 RIPs at levels above the -RT background; however, it was present in cDNA libraries made from total RNA. These data are discussed in the text in subsection “Dbp5 supports re-export of mature tRNAs following nutritional stress” and shown in Figure 5—figure supplement 1C.

One would have expected that L12A and R423A would have shown different levels of tRNA pulldown given their different roles as analyzed by FISH; how do the authors think about this.

As shown in Figure 5 and discussed in subsection “Dbp5 supports re-export of mature tRNAs following nutritional stress” and subsection “Dbp5 acts within the nucleus to support tRNA export” of the revised manuscript, L12A has a significant enrichment for binding tRNAIle. We interpret this increased interaction to be reflective of the increased nuclear pool of Dbp5. In the case of R423A, the interaction with unspliced tRNAs appeared reduced potentially corresponding with altered nuclear import, but this did not reach a significance level of p<0.05. Still, reduced binding is not the only means by which this mutation could impact tRNA export. For example, our data shows that RNA does not stimulate the ATPase activity of R423A to the level of wildtype Dbp5. As such, although binding events occur and are detected, binding may not be coupled with productive ATPase cycles that are required to support tRNA export. We have expanded the discussion of the R423A mutant in subsection “Dbp5 acts within the nucleus to support tRNA export” of the revised manuscript to reflect these comments.

4) Subsection “Gene expression in dbp5-L12A and dbp5-R423A at steady-state and in response to MMS”: no data are shown for the claim that steady state RNA levels were similar for wt, L12A and R423A mutants.

Through revision of the manuscript this statement has been removed.

5) Figure 4: it is nearly impossible to read the identities of the genes through the magenta and purple colors.

Colors have been updated to allow gene names to be more easily read. Please see current Figure 3 (previously Figure 4).

6) Figure 4A,B: differences in RNA levels in the mutants upon MMS treatment would be better shown as correlation plots. The data are hard to parse in their current form.

Through revision of the manuscript these data have been removed.

7) Subsection “Gene expression in dbp5-L12A and dbp5-R423A at steady-state and in response to MMS”: The FDR values for the highlighted GO terms are large (i.e. barely significant), raising the question whether the reported enrichments are biological meaningful. The authors make the point that L12A and R423A cause opposite changes in gene expression in the GO categories. Are the same genes affected? If so, this should be shown. If not, the FDR values for enrichment would probably render this effect statistically insignificant. More depth is needed in this section.

Through revision of the manuscript these data have been removed.

8) Subsection “dbp5-L12A and dbp5-R423A genetic interaction profiles”: The referenced Figure 4—figure supplement 1 seems to be missing.

Through revision of the manuscript these data have been removed.

[Editors' note: further revisions were requested prior to acceptance, as described below.]

The manuscript has been improved and acceptance is recommended in principle, but there is one remaining issue that needs to be addressed before acceptance, the new Figure 6 does not provide identification for all the proteins, either on the figure or in the legend.

We have revised the manuscript by Lari et al., entitled “*A nuclear role for the DEAD-box protein Dbp5 in tRNA export*” as requested. Specifically, we have added greater detail to the model presented in Figure 6 and the associated legend describing the figure.